# The 1430s: A period of extraordinary internal climate variability during the early Spörer Minimum and its impacts in Northwestern and Central Europe

Chantal Camenisch[1,2], Kathrin M. Keller[1,3], Melanie Salvisberg[1,2], Benjamin Amann[1,4,5,] Martin Bauch[6], Sandro Blumer[1,3], Rudolf Brázdil[7,8,] Stefan Brönnimann[1,4], Ulf Büntgen[1,8,9], Bruce M. S. Campbell[10], Laura Fernández-Donado[11], Dominik Fleitmann[12], Rüdiger Glaser[13], Fidel González-Rouco[11], Martin Grosjean[1,4], Richard C. Hoffmann[14], Heli Huhtamaa[1,2,15], Fortunat Joos[1,3], Andrea Kiss[16], Oldřich Kotyza[17], Flavio Lehner[18], Jürg Luterbacher[19,20], Nicolas Maughan[21], Raphael Neukom[1,4], Theresa Novy[22], Kathleen Pribyl[23], Christoph C. Raible[1,3], Dirk Riemann[13], Maximilian Schuh[24], Philip Slavin[25], Johannes P. Werner[26], Oliver Wetter[1,2]

[1]Oeschger Centre for Climate Change Research, University of Bern, Bern, Switzerland

[2]Economic, Social, and Environmental History, Institute of History, University of Bern, Bern, Switzerland

[3]Climate and Environmental Physics, Physics Institute, University of Bern, Bern, Switzerland

[4]Institute of Geography, University of Bern, Bern, Switzerland

[5]Department of Geography and Planning, Queen's University, Kingston (ON), Canada

[6]German Historical Institute in Rome, Rome, Italy

[7]Institute of Geography, Masaryk University, Brno, Czech Republic

[8]Global Change Research Institute, Czech Academy of Sciences, Brno, Czech Republic

[9]Swiss Federal Research Institute WSL, Birmensdorf, Switzerland

[10]School of the Natural and Built Environment, The Queen's University of Belfast, Northern Ireland

[11]Department of Astrophysics and Atmospheric Sciences, Institute of Geosciences (UCM-CSIC), University Complutense, Madrid, Spain

[12] Department of Archaeology and Centre for Past Climate Change, School of Archaeology, Geography and Environmental Science, University of Reading, Reading, UK

[13]Institute of Environmental Social Sciences and Geography, University of Freiburg, Germany

[14]Department of History, York University, Toronto, Canada

[15]Department of Geographical and Historical Studies, University of Eastern Finland, Joensuu, Finland

[16]Institute of Hydraulic Engineering and Water Resources Management, Vienna University of Technology, Vienna, Austria

[17]Regional Museum, Litoměřice, Czech Republic

[18]Climate & Global Dynamics Laboratory, National Center for Atmospheric Research, Boulder, USA

[19]Department of Geography, Climatology, Climate Dynamics and Climate Change, Justus Liebig University, Giessen, Germany

[20]Centre for International Development and Environmental Research, Justus Liebig University of Giessen, Giessen, Germany

[21]Aix-Marseille University, Marseille, France

[22]Johannes Gutenberg University of Mainz, Germany

[23]University of East Anglia, Norwich, UK

[24]Historisches Seminar and Heidelberg Center for the Environment, University of Heidelberg, Germany

[25]School of History, Rutherford College, University of Kent, Canterbury, UK

[26] Department of Earth Science and Bjerknes Centre of Climate Research, University of Bergen, Bergen, Norway

*Correspondence to*: C. Camenisch (chantal.camenisch@hist.unibe.ch)

Abstract. Throughout the last millennium, changes in the climate mean state affected human societies. While periods like the Maunder Minimum in solar activity in the 17[th] century have been assessed in greater detail, earlier cold periods such as the 15[th] century received much less attention due to the sparse information available. Based on new evidence from different sources ranging from proxy archives to model simulations, it is now possible to provide a systematic assessment about the climate state during an exceptionally cold period in the 15[th] century, the role of internal, unforced climate variability and external forcing in shaping these extreme climatic conditions, and the impacts on and responses of the medieval society in Northwestern and Central Europe. Climate reconstructions from a multitude of natural and anthropogenic archives indicate that the 1430s, a period coinciding with the early Spörer Minimum, was the coldest decade in Northwestern and Central Europe in the 15[th] century. The particularly cold winters and normal but wet summers resulted in a strong seasonal cycle in temperatures that challenged food production and led to increasing food prices, a subsistence crisis, and a famine in parts of Europe. As a consequence, authorities implemented numerous measures of supply policy in order to cope with the crisis. Adaptation measures such as the installation of grain storage capacities were taken by town authorities to be prepared for future events. The 15[th] century is characterised by a grand solar minimum and enhanced volcanic activity, which both imply a reduction of seasonality. Climate model simulations show that periods with cold winters and strong seasonality are associated with internal climate variability rather than external forcing. Accordingly, it is suggested that the reconstructed extreme climatic conditions during this decade occurred by chance and in relation to the partly chaotic, internal variability within the climate system.

## 1. Introduction

Several cold periods occurred in Europe during the last millennium and might have affected human socio-economic systems. The cold can be attributed to external climate forcing and internal (chaotic) climate variability. Forcing included cooling by sulphate aerosols from explosive volcanism and solar irradiance variations against the background of slow variations of Earth's orbit leading to a decrease in summer insolation over the past several millennia.

The past climate is reconstructed from information recorded in climate archives such as tree rings, sediments, speleothems, ice, and historical documents. Documented impacts of severe cold periods on socio-economic systems include reductions in the amount and quality of agricultural products. This in turn,

together with other political, social, and cultural factors, sometimes resulted in impacts on food availability and prices, famines, reductions in birth rates, population growth, population size, and social distress, many of which provoked adaptation policies and measures. While more recent cold events, such as the "Year Without Summer" after the 1815 eruption of Tambora (e.g., Luterbacher and Pfister, 2015) or the so-called Maunder Minimum in solar irradiation in the 17th century, are extensively discussed and documented in the literature (e.g., Eddy, 1976: Luterbacher et al., 2000, 2001; Xoplaki et al., 2001; Shindell et al., 2001; Brázdil et al., 2005; Yoshimori et al., 2005; Raible et al., 2007; Ammann et al., 2007; Keller et al., 2015), much less is known about an exceptionally cold period in Europe during the 15th century.

The aim of this study is to provide a systematic assessment of what is known about climate forcing, the role of internal, unforced climate variability, and socio-economic change during a particular cold period in Europe from around 1430–1440 CE (Fig. 1). This is done by exploring the output from simulations with comprehensive state-of-the-art climate models driven by solar and volcanic forcing, and by analysing multi-proxy evidence from various natural and anthropogenic archives to infer climate variability in terms of temperature, precipitation, and underlying mechanisms. Seasonality changes, which may have played an important role in generating impacts for medieval society, are discussed in detail. Historical documents are exploited to unravel socio-economic conditions, impacts, resilience, and adaption to change by using quantitative indicators such as corn prices, population and trade statistics, as well as descriptions. The potential impacts of climate on society are discussed in the context of other important socio-economic drivers.

Our study concentrates on Northwestern and Central Europe during the period of the Spörer Minimum (SPM) in solar activity in the wider context of the "Little Ice Age" (LIA; ~1300–1870). A particular focus is on the decade 1430–1440, which coincides with the early SPM. We stress that this temporal concurrence does not imply causality, and that the particular climatic conditions during the 1430s are not necessarily the result of changes in solar irradiation. Concerning the temporal extend of the SPM, a number of differing definitions exist: 1400–1510 (Eddy 1976b; Eddy 1977; Jiang and Xu, 1986); 1420–1570 (Eddy, 1976b; Eddy, 1977; Kappas 2009); 1460–1550 (Eddy, 1976a). Here, we use the years 1421–1550. The period of strongest reduction in incoming total solar irradiance (TSI) occurred during ~1460–1550 (Eddy, 1976a) and coincides with several large volcanic eruptions (Sigl et al., 2013; Bauch, 2016).

Historic documents show that the 1430s were a period of enhanced seasonality with cold winters and normal summers (Luterbacher et al., 2016; Fig S1). Yet, such changes in seasonality have not been assessed in detail using climate proxy data nor climate model output (Wanner et al., 2008). It remains unclear how, if at all, this seasonality was linked to external forcing or resulted by chance from internal climate variability and whether the seasonality was extraordinary in the context of the last millennium.

Concerning the hemispheric-scale mean changes climate model simulations and multi-proxy climate reconstructions agree in that the SPM was a period of rather cold conditions (e.g., Fernández-Donado et al.,

2013; Lehner et al., 2015). A recent study collecting hemispheric-scale reconstructions suggests a more diverse picture of temperature changes with different regions having opposed trends during the SPM (Neukom et al., 2014). Europe seems to have been only slightly cooler than average during the SPM (PAGES 2k consortium, 2013; Luterbacher et al., 2016). The authors state that only the Maunder Minimum

was a globally coherent cold phase during the last millennium. Recently, these continental-scale reconstructions are compared to the latest simulations of the Paleoclimate Modelling Intercomparison Project III (PMIP3; Schmidt et al., 2011), showing that models tend to overemphasise the coherence between the different regions during periods of strong external forcing (such as the SPM; PAGES2K-PMIP3 Community et al., 2015). Still, the simulations agree with the reconstructions for Europe in that a

major cooling happens after 1450, so after the 1430s.

In Western Europe, the 1430s featured a series of extremely cold and extended winters (Buismann, 2011; Camenisch, 2015b; Fagan, 2002; Lamb, 1982; Le Roy Ladurie, 2004), which affected the productivity of terrestrial ecosystems in the subsequent growing seasons. The consequences were losses in agricultural production. Crop failures, famines, epidemic plagues, and high mortality rates haunted large parts of

Europe at the end of this decade and in the 1440s (Jörg, 2008; Camenisch, 2012). Weather conditions during winter affected the food production and food prices in different ways (Walter, 2014; Camenisch, 2015b). An exceptionally cold and/or long winter can be the reason that, despite good growing conditions in the subsequent summer, terrestrial ecosystem productivity was substantially decreased (causes include cold injury, alterations of the energy and water balance, and advanced/retarded phenology; e.g., Williams et

al., 2015). For instance, very low temperatures could destroy the winter seed (mostly rye or wheat), which was sown in the fields in autumn. Usually, the winter temperatures do not have much influence on grain production, but in the case of the 1430s temperatures sank to such extremely low levels that – combined with no or almost no snow cover – the seedlings were damaged or destroyed (Camenisch 2015b; Pfister 1999). Late frosts, as occurred during the 1430s, usually had a devastating effect on grain production.

Additionally, cattle as well as fruit and nut trees suffered from very low temperatures. Frozen rivers and lakes could cause disturbances in the transport of food and consequently the food trade. Frozen bodies of water and drifting ice were also responsible for broken bridges and mills. The first meant disrupted trade routes and the latter interferences with regard to the grinding of grain into flour (Camenisch, 2015b). Thus, this period is an historic example of how society reacted to extreme climatic conditions and other changes

such as abruptly rising food prices, market failure, famine, epidemic diseases and wars and how adaptation strategies were implemented. Still, whether the famines associated with the documented crop failures were a mere result of climate change is questioned as prior to but also during the SPM international trade went through a period of deepening recession, hindering the people to sufficiently mitigate crop failure (Campbell, 2009; Jörg, 2008; Camenisch, 2015b).

Given this lack of understanding, it is timely to combine available evidence in a systematic study, from external forcing to climate change and implications to adaptation in an historical perspective. The outline is

as follows: section 2 focuses on the physical system during the SPM and presents climate reconstructions from different proxy archives. Section 3 presents climate model results and explores the role of external forcing versus internal variability. In section 4, socio-economic implications are analysed using historical evidence. Furthermore, this section illustrates how society reacted and which strategies were pursued in order to adapt. A discussion and conclusions are provided in the last section, which aims at stimulating a future focus on this period of dramatic impacts in Europe.

## 2. Reconstructions of climate during the Spörer Minimum

Sixteen comprehensive multiproxy multisite datasets covering Western and Central Europe are analysed to characterise the mean climate and seasonality during the SPM (Appendix, Fig. 2). The data include annual or near annual, well-calibrated, continuous series from tree rings, lake sediments, speleothems, and anthropogenic archives (see Table 1) covering the period 1300 to 1700. Summer temperature is represented by seven data series (Büntgen et al., 2006, 2011; Camenisch, 2015a; Riemann et al., 2015; Trachsel et al., 2010, 2012; van Engelen et al., 2001) and winter temperature by five data series (Camenisch, 2015a; de Jong et al., 2013; Glaser and Riemann, 2009; Hasenfratz et al., in preparation; van Engelen et al., 2001). Four data series provide information about summer precipitation (Amann et al., 2015; Büntgen et al., 2011; Camenisch, 2015a; Wilson et al., 2013).

In a first analysis, the centennial-scale variability is investigated by comparing the temperature mean of the SPM (1421–1550) with the preceding century (1300–1420) and the century afterwards (1550–1700). It appears that summer temperatures in Europe were not colder during the SPM than before or afterwards (not shown). On the contrary, the proxy series from Western Europe and the Swiss Alps that include lake sediment data and temperature reconstructions from chironomid transfer functions (Trachsel et al., 2010, 2012) reveal that, overall, the SPM was significantly ($p < 0.01$) warmer than the periods before and afterwards. For winter temperatures, a similar conclusion can be drawn from the reconstructions, i.e., the deviations were not unusual during the SPM in light of the early LIA.

While the centennial-scale climate variability informs mainly about the influence of the prolonged TSI minimum during the SPM, inter-annual to decadal-scale climate variability illustrates (cumulative) volcanic forcing or internal (unforced) variability. Fig. 2 shows the decadal means of the standardised proxy series. Decadal-scale variability shows pronounced temporal and spatial heterogeneity across Europe. Summers from 1421 to 1450 were consistently normal or warm (for the years 1430–1439, Luterbacher et al., 2016, see supporting information, Fig S1). Striking is the very cold decade 1451–1460, which is a consistent feature across all summer temperature proxy series and coincides with two consecutive very large volcanic eruptions in 1453 (unknown) and 1458 (Kuwae; Sigl et al., 2013). These cold summers across Europe persisted for one or two decades and were followed by rather warm summers until the 1530s, particularly in the Alps. Similar decadal-long cold summer spells were observed between 1590 and 1610, which also

coincided with two very large volcanic eruptions (Ruiz in 1594 and Huaynaputina in 1600; Sigl et al., 2013).

Winter temperature variability behaved differently. In Western Europe, the coldest conditions are reconstructed during the 1430s. The slightly warm anomaly on record [8] can be explained by its location in the Alps. Situated at 1791 m.a.s.l., during the winter the site is often decoupled from the boundary layer and, as such, is of limited representativeness for the lowlands. From 1450 to 1500, very strong winter cooling was observed in both the Alps and Poland. At least for these areas, consecutive strong volcanic forcing seemed to result in very cold and long winters (Schurer et al., 2014; Hernańdez-Almeida et al., 2015). Cold winters were also confirmed in these areas after 1590 (1594 Ruang and 1600 Huaynaputina eruptions; Arfeuille et al., 2014; Sigl et al., 2014).

A way to identify such years with high seasonality (i.e., cold winters and normal to warm summers) is the comparison of summer and winter temperature reconstructions. Fig. 2 shows that such a period was only evident from 1431–1440 in the proxy records. Additionally, the summer precipitation is shown in order to assess whether during this period the hydrological cycle was also unusual, either too dry or too wet, which may have enforced potential impacts due to a short growing season. Given the rather sparse information of only four records no consistent behaviour is found, i.e., some records show normal condition whereas one record show a strong increase in summer precipitation.

### 3. Modelling the climate state during the Spörer Minimum

For the 1430s, the reconstructions show an increase in seasonality: consistently normal or warm European summers coincide with very cold winters in Western Europe. Whether or not these changes in seasonality are due to external forcing or internal variability of the climate system cannot be answered by the reconstructions alone. Therefore, simulations with comprehensive climate models for the last millennium are analysed to identify underlying mechanisms and to discuss the relationship between reconstructed variability and external forcing factors (Schurer et al., 2014). Our ensemble of opportunity (see Table 2) includes simulations from the PMIP3 archive (Schmidt et al., 2011) as well as two newly provided transiently forced (HIST) and control (CNTRL; 600 years with perpetual 850 CE forcing) simulations using the Community Earth System model (CESM; Lehner et al., 2015; Keller et al., 2015).

The dominant forcing factors during the last millennium prior to 1850 were changes in solar activity and volcanic aerosols, with additional small contributions from changes in the Earth's orbit, in land use, and in greenhouse gas concentrations (Stocker et al., 2013). The total forcing applied to the different models, including solar, volcanic greenhouse gases, and anthropogenic aerosol contributions is shown in Fig. 3a. The largest inter-annual changes are due to volcanic forcing, despite large differences between models. A 31-yr moving average filtered version of the total forcing is shown in Fig. 3b, illustrating the contribution of volcanic forcing at inter-annual to multi-decadal time scales.

There are uncertainties in the climatic conditions simulated by the different models due to the use of different solar and volcanic forcing reconstructions in various models, how these forcings are implemented in a given model, as well as model-specific internal variability. The SPM features reduced solar irradiance and coincides with two dominant volcanic eruptions in 1453 and 1458 (Sigl et al., 2013; Bauch, 2016). The latter eruption, Kuwae, was previously dated to 1452/53 and appears at this date in the standard model forcings. As to the change in solar activity, most models include changes in TSI. However, the magnitude of the changes of TSI remain unknown and might be anywhere between 1 and several W/m$^2$ (e.g. Steinhilber et al., 2010; Shapiro et al., 2011). In addition, potential feedback mechanisms exist involving, e.g., stratospheric dynamics (e.g., Timmreck, 2012; Muthers et al., 2015).

The models analysed here simulate an average decrease in the temperature of the Northern Hemisphere from 1050–1079 to 1450–1479 of about 0.4°C, consistent with earlier studies (Fernández-Donado et al., 2013; Fernández-Donado et al., submitted). Miller et al. (2012) were able to simulate the LIA cooling due to volcanic eruptions alone, without invoking changes in solar activity. In their model, amplifying feedbacks involving a change in the North Atlantic ocean circulation cause a long-term cooling of the climate to the eruptions in the 13$^{th}$ and 15$^{th}$ century. Similarly, Lehner et al. (2013) found that a negative solar or volcanic forcing leads to an amplifying feedback also involving sea ice changes in the Nordic Seas.

While oceanic feedbacks following an initial volcanic or solar trigger mechanism might not be separable, the initial response of the European climate to volcanic and solar forcing is expected to be different in terms of its seasonality. Both forcings are expected to cool during summer, but while low solar forcing is expected to weaken the Westerlies and lead to low temperatures in Eastern Europe (e.g. Brugnara et al., 2013), volcanically perturbed winters tend to have a stronger westerly flow and higher temperatures in northeastern Europe (Robock, 2000). Note, however, that the mechanism of how changes in solar activity affect weather and climate is still not well understood and thus these mechanisms may not be implemented in climate models. The climate influence may proceed through changes in TSI, solar UV (Gray et al., 2010), or energetic particles (Andersson et al., 2014), which may have varying temporal developments. Further, reconstructions of the variations in solar radiation rely on proxy information such as sunspot counts or the abundance of radiocarborn and beryllium isotopes in tree rings or ice cores and are thus affected by uncertainties.

The modelled seasonality ($T_{JJA}$–$T_{DJF}$; for time series of both variables, see supporting information) of temperature in Europe is stronger in years with cold winters. This is illustrated by results from CESM (Fig. 4). The temperature difference between summer and winter is 13.06 ± 0.98 K averaged over Europe. The seasonality is increased to 14.27 ± 0.84 K when considering only years with very cold winters; here a winter is considered very cold if its temperature is within the lowest 17% of all winters. No such dependence can be found for precipitation. Overall, 56.8% of the years with a very large (above 1 standard deviation) seasonality coincide with a very cold winter. There is no difference between the control and the transient simulation concerning the occurrence of cold winters (HIST: 15.0% / CNTRL: 15.2% of all years)

as well as seasonalities, thus implying that, on average, external forcing does not affect modelled seasonality in Europe.

External forcing could also affect the seasonality during specific time periods. Based on winter temperatures, extremely cold decades are identified in all available simulations (see supporting information). However, the lack of consistency between models indicates that there is no clear link between external forcing and an increase in the occurrence of cold winter decades.

Maps of temperature and precipitation for the years with strong seasonality in temperature are given in Fig. 5, based on the transient simulation with CESM. In agreement with the reconstructions, years with strong seasonality show anomalously cold winters in Europe. The effect on the annual mean temperatures, however, is limited to certain regions; the reason is the partial cancellation of cold winters and warmer-than-average summers. Anomalies in precipitation also show large spatial differences. During winter, it is wetter than usual in Southern Europe and drier than usual in Western and Central Europe.

Volcanic eruptions are an important forcing factor, and since one of the strongest eruptions of the last millennium occurred within the SPM, a superposed epoch analysis is applied to the seasonality of temperature and precipitation in the multi-model ensemble. The superposed epoch analysis shows the mean anomaly of the 10 strongest volcanic eruptions with respect to the unperturbed mean of the five years before an eruption. As illustrated in Fig. 6 (for maps, see supporting information), after an eruption, the annual mean temperature is reduced over Central Europe whereas precipitation shows no signal. The seasonality of temperature shows a reduction in seasonality, especially in the year of an eruption. A volcanic eruption tends to induce an NAO-positive-phase-like pattern that eventually leads to a warming of Central Europe in winter while, during summer, the radiative cooling of the volcanic aerosols dominates. Precipitation seems to reflect the temperature behaviour, i.e., it mainly follows thermodynamics (Clausius-Clapeyron equation). Thus, the simulations suggest that in periods of frequent volcanic eruptions seasonality is reduced, in contrast to the increased seasonality in the 1430s decade. This also suggests that the exceptionally cold winters in this decade are not the result of volcanic forcing.

## 4. Climate and weather impacts on the economy and society during the early Spörer Minimum

Human societies are strongly influenced by climate, climate variability and extreme weather conditions (Winiwarter, Knoll 2007). These influences can be divided into short-term impacts (such as subsistence crises), conjunctural (price movement developments) and long-term impacts (e.g., decline of empires, big migration movements) (de Vries, 1980). Furthermore, in regard to a subsistence crisis as an example of a short-term climate impact, different levels of influence can be determined as the simplified climate-society-interaction model demonstrates (see Fig. 7). On the first level, primary production (food, feed, and fuelwood), water availability, and microorganisms are directly affected by weather conditions. Economic growth (through prices of biomass or energy) as well as epidemics and epizootics are in turn influenced by

these first-order impacts. The third level comprises demographic and social implications such as malnutrition, demographic growth, and social conflicts while cultural responses and coping strategies (e.g., religious rituals, cultural memory, learning processes, adaptation) constitute fourth-order impacts (Krämer, 2015).

This simplified climate-society-interaction model (see Fig. 7) gives the structure of how the climate impacts on society during the 1430s are presented in this paper, starting with a description and extreme weather conditions, followed by the description of the climate impacts on society, level by level. The respective information is available in a variety of historical documents such as narrative or administrative sources of different origins (Brázdil et al., 2005; Camenisch, 2015a; Bauch, 2016). Here, mainly
contemporary English, German, Hungarian, Czech, Austrian, Italian and Dutch charters, letters, manorial, town and toll accounts, as well as narratives are analysed.

The demographic, economic and political situation of Europe before and during the 1430s needs to be considered. Due to famine, the Black Death and repeated episodes of plague and other diseases Europe experienced a dramatic decline of population during the 14[th] century. During the first decades of the 15[th]
century the population stabilised but remained at very low levels. This did not change before the 1460s when European population began to grow again (Herlihy, 1987; Livi Bacci 1995, Campbell, 2016). As a consequence of the lower population pressure wages were rather high and living costs rather low in comparison to other periods. Furthermore, settlements were withdrawn from environmentally and politically marginal locations (Allen, 2001). Thus, the adverse effects of climate deterioration were offset
by the dwindling numbers of mouths to be fed and the shrinking proportion of households with incomes below the poverty line (Broadberry et al., 2015).

During the first half of the 15[th] century Europe suffered of the bullion famine, price deflation, major territorial and commercial losses to the Ottomans, and a sharp contraction in overseas trade were generating serious economic difficulties of their own (Day, 1987; Spufford, 1989; Hatcher, 1996). Several wars
aggravated the already tense situation. The food supply situation and the grain markets were influenced by them through several ways. Armies – confederates or enemies – marauded on the countryside in order to supply themselves. Furthermore, it belonged to the techniques of warfare of the time to weaken adversaries through destroying fields as well as seed and killing peasants and cattle. As a consequence, the rural populations sought refuge behind the walls of nearby towns, where the increasing demand for food led
explode the prices. In addition, wars led to increasing taxes, unsecure trade routes and a lack of farmworker and draught cattle when the territorial lord needed soldiers and horses for his military campaigns (Schmitz, 1968; Contamine et al., 1993; Camenisch 2015b).

In France, the Hundred Years War came into its last phase. In 1435, the Duke of Burgundy, a former ally of the English party, changed sides and again joined the French side. In the following 18 years, the English
party lost its entire territory on the continent with the exception of Calais. The recapture started during the

second half of the 1430s and included the devastation of parts of Flanders and Hainault by French troops. The desertion of the Duke of Burgundy in 1435 had further consequences for the economy of the Low Countries. The textile manufactories there were highly dependent on the import of English wool that failed for political reasons (Blockmans, 1980; Curry, 2012; Contamine et al., 1993; Derville, 2002). Furthermore,
the Low Countries had to pay high taxes for the maintenance of the Duke's armies involved in the recapture of the English territories in France. As a consequence, a number of cities in the Low Countries were in open rebellion against their Duke (Barron, 1998; van der Wee, 1978). Further to the East, the Czech Lands and the northern parts of the Hungarian kingdom in the early 1430s were still affected by the repercussions of the protracted Hussite wars (Brázdil and Kotyza, 1995). In the winter of 1431, the
Hungarian army – greatly fearing a Turkish attack – had increased its operations at the southern borderline due to the deeply frozen Danube (Hungarian National Archives, DL 54734). In Bologna, Italy, military actions and social unrest had weakened the city and its hinterland. Furthermore, communities in the contado complained about ravaging floods and claimed a reduction of their taxes towards the municipal authorities. Additionally, a serious earthquake hit the city simultaneously with the incessant rain and
worsened the situation (Bauch, 2015). The area of the Swiss confederation was also impacted by political troubles in the years preceding the Old Zürich War (1440–1446). This conflict about the possession of territories and hegemony in the area of today's Eastern Switzerland was fought out between the canton of Zurich and the cantons of Schwyz and Glarus together with the other cantons of the Old Swiss Confederacy. (Reinhardt, 2013; Maissen, 2010).

As the reconstructions in Sect. 2 (see Fig. 2) show, the weather conditions during the 1430s stood out due to harsh and chilly winters. In the historical sources many descriptions can be found. In the area of the Low countries the winters of 1431/32, 1432/33, 1434/35 and 1436/37 were extremely cold whereas the winters of 1433/34 and 1437/38 were very cold. In the same area spring temperatures were very low or extremely low in 1432, 1433 and 1435 (Camenisch 2015a). Bohemia, Austria, and the Hungarian Kingdom suffered
from a number of cold winters during the 1430s, especially the winter of 1431/32, 1432/33, 1434/35 were outstanding cold in these areas (Brázdil et al., 2006). These remarkably cold winters caused the freezing of rivers and lakes in Central Europe, England, and the Netherlands and were accompanied by recurrent frost periods in April and May (Fejér, 1843; Marx, 2003; Brunner, 2004; Camenisch, 2015b). In Scotland during the winter 1432/33, for instance, the wine in bottles had to be melted with fire before it could be drunk.[1]
Extremely cold winters during the 1430s were also reported in Ireland (Dawson, 2009). In South-eastern France the winter seasons from 1434 until 1437 were outstandingly cold. In addition, there were frost periods in April 1432 and 1434 mentioned in that area (Maughan, 2016). In North and Central Italy, the

---

[1]

"In the time of King James the First […] a vehement frost was in the winter afore, that wine and ail was sauld be pound wechtis and meltit agane be the fire." (Dawson, 2009: 106)

winter of 1431/1432 was extremely cold till April 1432 (Bauch, 2015). In addition, in the Low Countries the summer seasons of 1436 and 1438 were also very cold (Camenisch 2015a).

In South-eastern France, in the Provence area, and in the Netherland the first half of the 15[th] century was characterised by high levels of hydro-climatic variability. From 1424 to 1433 two flood and five drought years occurred (Pichard and Roucaute 2014; Glaser and Stangl, 2003). South of the Alps, the time span from 1430 to 1433 was extraordinarily wet (Bauch 2015). Likewise, during the 1430s, Bohemia, Austria, and the Hungarian Kingdom suffered from one of the greatest known flood anomalies characterised, for example, by the 'millennial' July 1432 flood in Bohemia (Brázdil et al., 2006) or by the significant floods of the Danube reported in 1432, 1433, 1436, 1437, 1439, and 1440 (see e.g., Brázdil and Kotyza, 1995; Rohr, 2007; Kiss, 2012). Major flood events and their consequences were also documented in the second half of the decade (e.g., in 1435, 1437, 1438 and 1440) in the eastern part of the Carpathian Basin, in Transylvania, and in the Tisza catchment (Brázdil and Kotyza, 1995; Rohr, 2007; Kiss, 2011). In the Low Countries the summer seasons of 1432 and 1438 were very wet (Camenisch 2015a). An analogical temperature and precipitation pattern is also indicated by CESM (see Fig. 5).

The main first-order impact during these years was a decline in food production. In England, Germany, France, the Netherlands, Bohemia, and other places, crop failures were reported in 1432, 1433, 1434, 1436, 1437 and 1438 (Jörg, 2008; Tits-Dieuaide, 1975; Camenisch, 2012, Brázdil et al., 2006). In late April 1434, vineyards were damaged by frost in Hungary, Austria, and Bohemia. In Italy, the years 1431–1435 were characterised by harvest failures and dearth (Bauch, 2015). During the harsh winters of 1434/35 and 1436/37, in the London area special references were made to herbs such as laurel, sage, and thyme, which were destroyed by the frost. Moreover, the lack of fire wood and coal is mentioned (Brie, 1906a). In the area of the Low Countries and the Holy Roman Empire, several authors describe frozen vineyards, devastated winter grain, and damages to livestock during the winter of 1436/37. Vegetables, vine, and grain in the fields were destroyed by two frost periods at the end of March 1437 and in the second half of May (Camenisch, 2015b). Harvest failures and grain shortages were also mentioned in the area of Berne in the same year (Morgenthaler, 1921). In 1440, serious losses in wine production and a bad hay harvest were reported for Pozsony/Pressburg (which is todays Bratislava) (Ortvay, 1900).

As a consequence of the poor harvests in many European regions, food prices increased considerably (second-order impact according to Krämer's model, see Fig. 7). Early reports on rising food and firewood prices in Paris, Cologne, Augsburg, and Magdeburg date back to the years 1432 and 1433 (Beaune, 1990; Cardauns et al., 1876). In 1433, high food prices prevailed in Austria, the Czech Lands, and the Hungarian kingdom (Höfler, 1865). Even in Scotland and Ireland, high prices and shortages were mentioned in the same year (Dawson, 2009). Special attention was paid to the price development of eatables in 1437/38 and 1438/39 in London (Brie, 1906a). In many other places in the Holy Roman Empire and the Low Countries, very high food prices were mentioned in the second half of the 1430s (Jörg, 2008; Camenisch, 2015b). In England, the situation seemed more complicated. A chronicle reported increasing wheat prices in 1435 and

the consumption of substitute food such as bread made from fern roots was reported in the North (Marx, 2003). In London, rising prices for different grains were noted as well as for wine, sweet wine, meat, and fish. The consequences that were described for the wider population were inferior bread and malnutrition (Brie, 1906a). Other sources proved moderate prices in 1435 and no price increases in England before 1438 (Munro, 2006).

In almost all historical sources which have been examined for this research epidemic diseases are mentioned simultaneously with cold and wet weather conditions, dearth and subsistence crisis. Yet, often it is not possible to identify the type disease since an exact description is lacking and most diseases were just called "pestis". Several links between weather conditions and diseases are known. Cold and humid weather favour the spread of certain diseases of the respiratory system (Litzenburger, 2015). Also ergotism – then called Saint Anthony's fire and perceived as an epidemic disease and not as a dangerous and potentially lethal intoxication through the ergot fungus as it actually is – is linked to cold and humid weather. The fungus prospers best in a humid and rather cold environment (Billen, 2010). The relationship between weather conditions and the plague is still part of an ongoing discussion (Audouin-Rouzeau, 2003; Saluzzo, 2004). Furthermore, undernourished people were prone to diseases of the digestive and respiratory system and infections (Galloway, 1988; Landsteiner, 2005; Campbell, 2009).

Diseases resurged in these years and deaths from the plague were widely reported during the serious famine of 1438/1439, when predisposing environmental and economic conditions favoured host-vector-human interactions, and from 1450–1457, when summer temperatures were the most depressed and ecological stress was again acute (Biraben, 1975). Epidemics and high death rates were mentioned in the North of England (Brie, 1906a). Furthermore, 'pestilentia' was reported as far east as the Hungarian kingdom (e.g. ca. 1430: Iványi, 1910; 1440: Hungarian National Archives DL 55213). During the second half of the 1430s, Italy saw a row of country-wide epidemics (Bauch, 2015). In Bruges 24000 death people due to epidemics and famine were mentioned (Camenisch, 2015b). Around Easter of 1439, the epidemic disease also reached Berne where a considerable part of the town's inhabitants was carried off (Morgenthaler, 1921). During the 1440s and 1450s, Europe's population sank to its lowest levels during the Late Middle Ages, due to epidemiological and reproduction regimes that kept deaths in excess of births (McEvedy and Jones, 1978; Broadberry et al., 2015).

It also appears that the extreme weather of the 1430s had a strong impact on the health and fertility of sheep flocks in England. Thus, as several manorial accounts from south English demesnes reveals, the years 1432, 1433, 1437 and 1438 saw excessive mortality rates in sheep flocks, with the average figures standing at 32 per cent (compared with 4–5 per cent in normal years). The weather seems to have also affected the fertility rates of ewes (calculated as the ratio between newborn lambs and all mature female sheep). The figures stood at 83 in 1434 and fell to about 55 in 1437 and 1438 (East Sussex Record Office, S-G/44/85-94). It should be borne in mind that in the late-medieval period, about 90–95 lambs were expected to be born of 100 ewes in normal years. The decline in sheep health and fertility rates also implied a decline in

the productivity rates of sheep. In the 1430s, the average fleece weight per mature sheep was 1.1 lbs, falling to 1 lb in the 1440s and the 1450s (compared with the average of 1.4 lbs for the period 1210–1455) (Stephenson, 1988). The fall in wool productivity is reflected in the annual export levels of English wool, which fell from an annual 13,359 sacks (each sack = 364 lbs) in 1426–1430 to 9,385 sacks in 1431–1435 and 5,379 sacks in 1436–1440. The respective figures for 1437, 1438, and 1439 were 1,637; 1,548; and 1,576 sacks a year (Carus-Wilson and Coleman, 1963).

The impact of the extreme weather on the health of other animals is less clear. In 1434–1435, 37% of all cows died at Alciston (Sussex), but this seems to have been a local, rather than national outbreak. Also, the fertility rates of cows declined from about 90 to 66 in that year, on the same demesne (East Sussex Record Office, S-G/44/85-94). More detailed research is needed, in order to determine to what extent the situation at Alciston is reflective of other parts of England.

As has been shown, food shortages and crisis are mentioned at many places during the 1430s in North-western and Central Europe. Most places were already affected during the first part of the 1430s as has been shown in regard to the rising prices. In the years from 1432–1434 Bohemia was confronted with famine. During the second part of the decade especially the Low Countries and the Holy Roman Empire suffered a veritable famine. The author of the *Tielse kroniek* described the year 1438 with the following words: In 1438 there was such a dearness and famine in the entire Netherlands so that one did not know how to complain about poverty and moan on misery.[2] Almost everywhere people tried to cope with the dearth.

Usually, grain was traded whenever the price difference between two places was high enough to yield a profit despite the high transport costs; this was rather often the case. During the 15$^{th}$ century grain trade occurred regularly in many European regions (Achilles, 1959; Camenisch, 2015b). Grain was bought from distant places in order to increase the food offerings and consequently stabilise food prices and supply people with victuals. In London, Mayor Stephen Brown organised the successful import of rye from Prussia (Brie, 1906b). Thus, the narrative sources written for the nobility and the merchant elite both living in London completely neglect the effects of the granaries erected during the 14$^{th}$ and 15$^{th}$ century (Grandsen, 1982; Keene, 2012). In Great Yarmouth, a seaport in Norfolk with a focus on herring fishery and trade, grain was usually used to fill up the ships to maximise profits on the return journey. However, when harvests failed in northern and central Europe due to poor weather in the late 1420s and especially from 1437–1439, Yarmouth's trade pattern changed completely. Merchants from the Low Countries were

---

«In 1438 heerste er zo'n duurte en zulk een hondersnood in geheel Neder-Duitsland dat men van armoede en ellende niet meer wist hoe te jammeren en te klagen.» (Kuys et al. 1983: 167)

purchasing large amounts of and sometimes exclusively grain to bring to the famished cities on the southern side of the North Sea. Due to extremely high grain prices, the long distance grain trade became so profitable that intermediary traders from the Thames estuary region organised large-scale shipments into the usually exporting Norfolk area, most likely to Norwich. To stop the flow of grain to the Low Countries, the English crown issued an export ban in September 1438, thereby closing England as a supply source to the Low Countries. During the remaining crisis years, the official records show very little grain leaving Norfolk via Yarmouth, and this grain was mainly sent to a number of small harbours along the East Anglian coastline. Smuggling across the North Sea was likely, but naturally not mentioned, in the customs rolls (Norfolk Record Office, Great Yarmouth Borough Archives, Court Rolls, Y/C 4/134-149; Calendar of the Close Rolls Henry VI). In (West-) Hungary, the food shortage was already a problem in 1433 due to the high volume of cereal exported to the neighbouring countries. Thus, in October 1434, a royal charter prohibited cereal export in order to avoid a great famine (Fejér, 1843). Such export bans were also established in the Low Countries and in the territory of the Teutonic Order in the Baltic area (Tits-Dieuaide 1975). In 1437 in the area of modern Switzerland, after a poor harvest, the town of Zurich excluded Schwyz and Glarus from the grain markets in its territory (Schnyder, 1937). This exclusion was a catastrophe since the cattle-breeding cantons of Schwyz and Glarus were dependent on these markets even in times of plenty; in times of dearth it was a deadly threat. After this embargo, Schwyz, Glarus, and their allies took up arms and began a war – the Old Zurich War – that lasted several years (Reinhardt, 2013; Maissen, 2010). Furthermore, at several places in the Holy Roman Empire the beer brewing was regulated during the years 1434 and 1437/38 (Jürg, 2008).

Mainly as a result of money devaluation (new silver coins: 1436) and taxation problems, one of the most significant medieval peasant uprisings occurred in 1437/38 in Transylvania; similar problems and a power-controversy between German and Hungarian citizens motivated the turbulence of the Buda inhabitants in 1439. In 1440, serious problems in wine production, bad hay, and poor cereal harvest formed the basis for a (royal) tax reduction in Pozsony/Pressburg (see e.g. Engel, 2001).

Also, religious responses to the bad weather conditions during the 1430s are known. In Bologna, the civic cult of the Madonna di San Luca started in 1433 as a reaction to the continuous rainfall from April to June of that year. The veneration of a miraculous icon was repeated one year later as bad weather returned; in the following decades, processions were organised when all kind of perils (like epidemics and war) threatened the civic community. With this approach to coping with this crisis, Bologna clearly followed the model of neighbouring Florence, where the Madonna dell'Impruneta was famous for helping the city in all kinds of (natural) disasters since 1333 (Bauch, 2015).

In several parts of the Holy Roman Empire, people blamed minorities for their misery. The perception and treatment of the Romani which were then called "gypsies" at the beginning of the 15[th] century is directly connected to the worsening of the weather during the early SPM. In chronicles of the 15[th] and 16[th] centuries, this connection was described as purely negative (Gronemeyer, 1987). For instance, the newly

arrived Romani were blamed for the worsening of the weather conditions during the years 1430 to 1440 as well as the associated consequences, including rising food prices, famine, and plagues (Winstedt, 1932). The ability to change or create weather was attributed to the 'gypsies' magical powers. The discrimination and persecution of the 'gypsies', especially in connection to misfortunes, could be seen as an attempt to solve underlying social tensions and problems. Climate change, in particular, entailed a variety of social problems through the shortage of resources. Thus, the statement that the newly arrived 'gypsies' were the cause for the worsening of the weather is an expression of this coping strategy. Furthermore, Jews were blamed for usury during the 1430s. In many towns of the Holy Roman Empire the Jews were expelled. The reasons for that were complex and are strongly linked to political reasons in regard to the Holy Roman Empire and the Church Councils in the first half of the 15[th] century. The tensions through the subsistence crisis only aggravated the situation (Jörg, 2013). During the following centuries the accusations of Jews squeezing profit from the misery of people which suffered from the consequences of subsistence crises by committing usury, hoarding of staple food for later profit and debasing of money did not vanish (Bell, 2008). However, in the course of the 15[th], 16[th] and 17[th] century 'witches' were suspected of 'weather-making'. They had the function of scapegoats in many cases of extreme weather events (Behringer, 1999; Pfister, 2007; Litzenburger, 2015).

As a consequence of the crisis of the 1430s communal granaries were built during the subsequent years at several places in Europe, for instance in Basel, Strassbourg, Cologne or London (Jörg, 2008; Dirlmeier, 1988; Campbell, 2009, Litzenburger, 2015). These building activities of the towns were an adaptation strategy that should prevent the society there from further food shortages.

Another example of how the climate during the SPM affected human society concerns fishery. Historical evidence plausibly connects the output of medieval fisheries for herring (*Clupea harengus*) in the North Sea and the Baltic to decadal-scale fluctuations in regional weather conditions. Preserved herring were the most important and widely-marketed fish product in Europe. In particular, they provided the cheapest protein-rich food permitted during the six weeks of Lent in late winter and early spring when Christian rules most harshly forbade consumption of animal products. Recent fisheries science has established a close relationship between the regional climate and the success of these herring stocks. Limits of herring ranges move northwards in warmer and southwards in colder decades. Furthermore, larval herring experience high mortality during cold late winters and springs in their primary habitat of the eastern North Sea, resulting in low adult populations and poor fishing 2–4 years later (Alheit and Hagen 1997, 2001, 2002; Archipelago Research Institute, 2015; Bailey and Steele, 1992; Finney et al., 2010; Krovnin and Rodionov, 1992; Poulsen, 2008).

Between the 1360s and 1540s, three kinds of historical sources indicate fluctuations in herring catches and stocks: contemporaries report losses in specific fisheries; 25 price series from 11 locations document great local volatility but also periods of widespread price peaks; and a unique record of yearly landings, written between 1405/06–1491/92 at Dieppe, a modest port near the southern boundary of the fishery range for

herring. Taken together, these records identify at least regional and temporary collapses of herring catches for a time after 1360 (in the southern North Sea), locally in the Øresund from the 1410s, and more generally during the 1440s–1460s, the 1480s, and 1520s–early 1530s (e.g. Allen-Unger Database, 2015; Gemmill and Mayhew, 1995; Gerhard and Engel, 2006; Hauschild, 1973; Hitzbleck, 1971; Rogers, 1866–
1902; van der Wee, 1963; Hoffmann, forthcoming). While some herring fisheries may have diminished in the 1430s, regional and general failures of catches and stocks were most likely greater during the cold spells of the 1360s, mid–1400s, and after 1520.

In the decades prior to the examined period, societies were less vulnerable to such large-scale famines. The reason for this lies in the described lower demographic pressure after the Great Famine and the Black Death at the beginning and in the middle of the 14th century, the higher wages and generally lower living costs
(Campbell, 2009). But why did Europe suffer from this crisis if the society was less vulnerable? This question cannot yet be answered completely. The crisis during the 1430s was the most severe since the beginning of the 14th century (Bauernfeind, 1993; van der Wee, 1978). This was probably the reason why societies and especially the authorities were not prepared to cope with failing markets and interrupted trade routes in such an extent that happened during these years (Jörg, 2008). Presumably, the high food prices
during the first part of the 1430s had the effect that food stocks were already consumed and there was no possibility to accumulate supplies during this period of poor harvests.

However, Europe was affected by this crisis from the Iberian Peninsula to the Baltic and the Russian Principalities as well as to the British Islands. Yet, there were big differences in the magnitude and
sequence of the crisis (Jörg, 2008; Contamine et al., 1993). In northern France, for instance, the famine formed the nadir of agrarian production and the demographic development of the 15th century (Neveux, 1975). In the Low Countries and the Holy Roman Empire, the crisis developed into a veritable famine. In other parts of Europe, the rising food prices did not result into famine; then an aggregate decline of 13 per cent in GDP per head in Italy, Holland, and England between 1435 and 1442, initiated by disastrous grain,
wine, and wool harvests, did not escalate into a demographic repeat of the Great European Famine of 1315–1321 (see Fig. 8; Malanima, 2011; van Zanden and van Leeuwen, 2012; Broadberry et al., 2015; Campbell, 2009). Clearly, too, there was little prospect of breaking out of the prevailing economic and demographic stagnation while the agricultural output remained depressed and harvests uncertain. Together, prevailing environmental and economic constraints were too strong (Campbell, 2012). It is not until the
final quarter of the 15[th] century, once the early Spörer Solar Minimum was past its worst, that incipient signs of regrowth become apparent in Italy, Spain, and England and especially in commercially enterprising Portugal and Holland (Campbell, 2013). Nonetheless, in many parts of Europe the next subsistence crisis did not occur before the 1480s (Morgenthaler, 1921; van Schaïk, 2013; Camenisch, in press).

The reason why some regions were hit more than others is difficult to detect. It can be assumed that wars and riots played an important role. Furthermore, the different levels of market integration, the unequal

dependence on the markets in order to feed the population and the demographic structure of the different regions are certainly of importance. Still, this is not yet sufficient in order to explain the magnitude of the crisis and the regional differences. Perhaps, institutional factors such as poorly conceived famine relief, lower tax base due to the declined population or higher transport costs as a consequence of the high wages

need to be examined in future research in order to better understand this crisis of the 1430s.

## 5. Conclusions

Here we have presented the first systematic assessment of the 1430s, a particularly cold period in Europe coinciding with the early Spörer Minimum in solar irradiation, characterised by devastating losses in agricultural production and the associated socio-economic consequences.

Natural (tree rings, lake sediments, and speleothems) and anthropogenic archives agree that the 1430s were subjected to very cold winters and normal to warm summers. This strong increase in the seasonality of temperature suggests that, despite normal climatic conditions in the growing seasons, terrestrial ecosystem productivity was substantially decreased during this decade.

State-of-the-art climate models indicate that this stronger seasonality was likely caused by internal natural

variability in the climate system rather than external forcing. In fact, the results suggest that strong volcanic eruptions decrease the seasonality of temperature and thus cause the opposite effect. Taken together, these lines of evidence indicate that the increased occurrence of extremely cold winters during this decade can be attributed to unforced, internal variability and the resulting atmospheric conditions.

In response to the prevailing weather conditions, harvest failures all over Europe were reported. These

harvest failures, together with other socio-economic factors, led to an increase in food prices. In particular, wars in different parts of Europe and market failures caused by export stops and other interruptions of trade played a role. Especially in the Low Countries, parts of France, and parts of the Holy Roman Empire, increasing food prices resulted in a subsistence crisis. Many coping strategies – implemented by civil or religious authorities as well as by the population itself – are documented during this crisis, including trade

regulations and restrictions on the brewing of beer. In the context of the crisis the Romani were blamed the first time for adverse weather conditions, rising food prices, famine and plague. Furthermore, the subsistence crisis was the reason for the subsequent construction of granaries in different towns in Europe. Until now little considered and analysed, this period provides a rich source of knowledge on how society reacted to deteriorating climate conditions, i.e., a shortening of the growing season due to a series of cold

winters and the associated increase in seasonality. This period demonstrates how different environmental and social factors and the interplay between them can generate strong impacts on the socio-economic system with consequences such as famine.

The 1430s are an outstanding period due to the fact that before those crisis years no supra-regional famine occurred since the middle of the 14[th] century. Although the extent of the crisis was a new experience for the societies, town authorities all over Europe started to implement supply policies or other coping strategies as has been shown. It should last another 40 years before the next subsistence crisis hit larger parts of Europe.

**Appendix: Data base of reconstructions**

A comprehensive set of paleoclimate records is considered in section 2 to provide a wide range of climate variables from different paleoclimate archives, which are representative for most of Northwestern and Central Europe (Fig. 2). Information about summer and winter temperatures as well as summer precipitation are obtained from historical sources, tree rings, speleothems, and varved lake sediments in
order to characterise the climate during the SPM (Tab. 1).

The datasets are selected according to the following criteria:

- Calibrated and validated proxy-climate relationship (demonstrated plausible mechanistic relation to climate);
- annual to near-annual resolution; covering the SPM (here: 1421–1550), and ideally the period
1300–1700;
- continuous, with no major data gaps; and
- published in a peer-reviewed journal (except Hasenfratz et al., in preparation).

All datasets are analysed at decadal-scale resolution. For comparability, all annual data are standardised with reference to the period period 1300–1700 (data sets 6, 11 and 16: 1400–1500). Finally, decadal means
(10-year mean windows) are calculated for each dataset.

**Acknowledgements**

This study is an outcome of the workshop "The Coldest Decade of the Millennium? The Spörer Minimum, the Climate during the 1430s, and its Economic, Social and Cultural Impact", organized by the three lead authors, which took place from 4 to 5 December 2014 at the University of Bern in Bern, Switzerland. The
workshop was supported by the Oeschger Centre of Climate Change Research (OCCR), the Intermediate Staff Association of the University of Bern (MVUB) and the Faculty of Humanities of the University of Bern. K.M.K., S. Blumer, F.J. and C.C.R. acknowledge support from the Swiss National Science Foundation through project 200020_159563. Simulations with NCAR CESM1 were carried out at the Swiss National Supercomputing Centre in Lugano, Switzerland.

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

## Tables

Tab. 1: Comprehensive multiproxy multisite reconstructions from Western and Central Europe representing summer and winter temperatures as well as summer precipitation.

| N° | Variable | Archive | Area | References |
|---|---|---|---|---|
| **Summer temperature** | | | | |
| 1 | JJA$_{anomaly}$ | Lake sediments | Switzerland / W-C Europe | Trachsel et al. (2010) |
| 2 | JJAS$_{anomaly}$ | Tree rings | Greater Alps | Büntgen et al. (2006) |
| 3 | JJA$_{anomaly}$ | Tree rings + lake sediments | Switzerland-Austria / W-C Europe | Trachsel et al. (2012) |
| 4 | JJA$_{anomaly}$ | Tree rings | W-C Europa | Büntgen et al. (2011) |
| 5 | JJA indices | Historical documents | C Europe | www.tambora.org / Riemann et al. (2015) |
| 6 | JJA indices | Historical documents | Low Countries | Camenisch (2015b) |
| 7 | MJJAS indices | Historical documents | Belgium-Netherlands-Luxemburg | van Engelen et al. (2001) |
| **Winter temperature** | | | | |
| 8 | ONDJFMAM | Lake sediments | Switzerland /W-C Europe | de Jong et al. (2013) |
| 9 | DJF indices | Historical documents | www.tambora.org / C Europe | Riemann et al. (2015) |
| 10 | DJF indices | Historical documents | Low Countries | Camenisch (2015b) |
| 11 | NDJFM indices | Historical documents | Belgium-Netherlands-Luxemburg | van Engelen et al. (2001) |
| 12 | Mean AWS Temp | Speleothems | Switzerland / W-C Europe | Hasenfratz et al., in preparation |
| **Summer precipitation** | | | | |
| 13 | MJJA | Lake sediments | Swiss Alps / W Europe | Amann et al. (2015) |
| 14 | AMJ | Tree rings | W-C Europe | Büntgen et al. (2011) |
| 15 | JJA | Historical documents | Low Countries | Camenisch (2015b) |
| 16 | MAMJJ | Tree rings | S-C England / W Europe | Wilson et al. (2013) |

5   Tab. 2: Overview of the climate models used in this study. Details of the respectively applied forcing can be found in Bothe et al., 2013; Lehner et al., 2015; PAGES2k-PMIP3 group, 2015, and references therein.

| Model (abbreviation) | Institute | Reference |
|---|---|---|
| CCSM4 | National Center for Atmospheric Research | Landrum et al. (2012) |
| CESM1 | National Center for Atmospheric Research | Lehner et al. (2015), Keller et al. (2015) |
| FGOALS-gl | State Key Laboratory of Numerical Modeling for Atmospheric Sciences and Geophysical Fluid Dynamics, Institute of Atmospheric Physics, Chinese Academy of Sciences | Dong et al. (2014) |
| FGOALS-s2 | | Bao et al. (2013) |
| GISS-E2-R | National Aeronautic and Space Administration, Goddard Institute for Space Studies | Schmidt et al. (2014) |
| IPSL-CM5A-LR | Institut Pierre-Simon Laplace des sciences de l'environnement | Dufresne et al. (2013) |
| MPI-ESM-P | Max Planck Institute for Meteorology | Jungclaus et al. (2014) |

**Figures**

Fig. 1: Illustration of the research disciplines and methods brought together in this systematic assessment.

Fig. 2: Individual paleoclimate reconstructions for summer temperature, winter temperature and summer precipitation. [Left] Dots are specific sites considered by the different authors (listed from 1 to 16; Table 1). [Right] Decadal-scale (10-yr mean) summer temperature, winter temperature and summer precipitation for the 16 climate reconstructions, standardized with reference to the period 1300–1700 (data sets 6, 11 and 16: 1400–1500). The black lines enclose the decade 1430–1440. References: [1]Trachsel et al. (2010), [2]Büntgen et al. (2006), [3]Trachsel et al. (2012), [4,14]Büntgen et al. (2011), [5,9]Riemann et al. (2015), [6,10,15]Camenisch (2015b), [7,11]van Engelen et al. (2001), [8]de Jong et al. (2013), [12]Hasenfratz et al., in preparation, [13]Amann et al. (2015), [16]Wilson et al. (2013).

Fig. 3: (a) Estimations of total external forcings used by the models in Tab. 2 according to Fernández-Donado et al. (2013). The panel includes anomalies with respect to the period 1500–1850 including the contributions of anthropogenic (greenhouse gases and aerosols) and natural (solar variability and volcanic aerosols). (b) 31-year moving average filter outputs of (a).

Fig. 4: Probability density functions of seasonality in surface temperature (TS; °C) averaged over Europe (8°W–22°E, 41–55°N) for all years and years with a very cold winter (conditional). The latter are defined by winter temperatures cooler than mean-1sigma. Left: for the years 850–1849 of the transient CESM simulation (HIST), right: also for the unforced control simulation (CNTRL; 600 yrs).

Fig. 5: Maps of surface temperature (top; °C) and precipitation (below; mm/day). Shown are annual (left), DJF (middle) and JJA (right) averages based on the transient simulation (years 850–1849) with CESM. First row: mean for all years. Second row: anomalies for years with strong seasonality in TS (> mean + 1sigma) compared to all years. Seasonality is defined as the difference between the means June–August and December–January of the respective year.

Fig. 6: Superposed Epoch Analysis on the ten strongest volcanic eruptions in six PMIP3 models, by measure of the respective forcing data set unit (aerosol optical depth or injection amount), over the period 850–1849. (Left) Annual mean temperature (top) and precipitation (bottom) and (right) seasonality defined as the difference between the means June–August and December–January of the respective year. Each of the 60 time series (6 models x 10 eruptions) is expressed as an anomaly to the mean of the five years preceding the eruption year (year 0). The shading indicates the 10–90% confidence interval, while the black solid line is the mean across all 60 time series. The red circles indicate significantly different means at 95% and 90% confidence according to a t-test comparing each year to the 5 year mean preceding the eruption year.

Fig. 7: This model developed by Daniel Krämer and Christian Pfister shows how climate interacts with society. Extreme weather causes biophysical effects on the first level, which can be followed by second order impacts that concern economic growth as well as human and animal health. On a third level are demographic and social implications situated whereas cultural responses act as fourth level impacts (Krämer, 2015; Luterbacher and Pfister, 2015).

Fig. 8: Crop yields from Southern England (wheat, barley, oats), Durham tithes, English grain prices and English salt prices for the years 1420–1460 (values are given as anomalies with reference to (w.r.t.) the period 1400-1479). Shown are the years 1437 and 1438 in the back-to-back grain harvest failure in South England, the massive reduction in Durham tithe receipts and the marked inflation of grain prices for three consecutive years. In 1442, the harvests in South England are again poor. As far as the agricultural impacts of the Spörer Minimum are concerned in England, 1432–1442 stands out as the worst period, especially 1436–1438 (adapted from Campbell, 2012).

# Figures

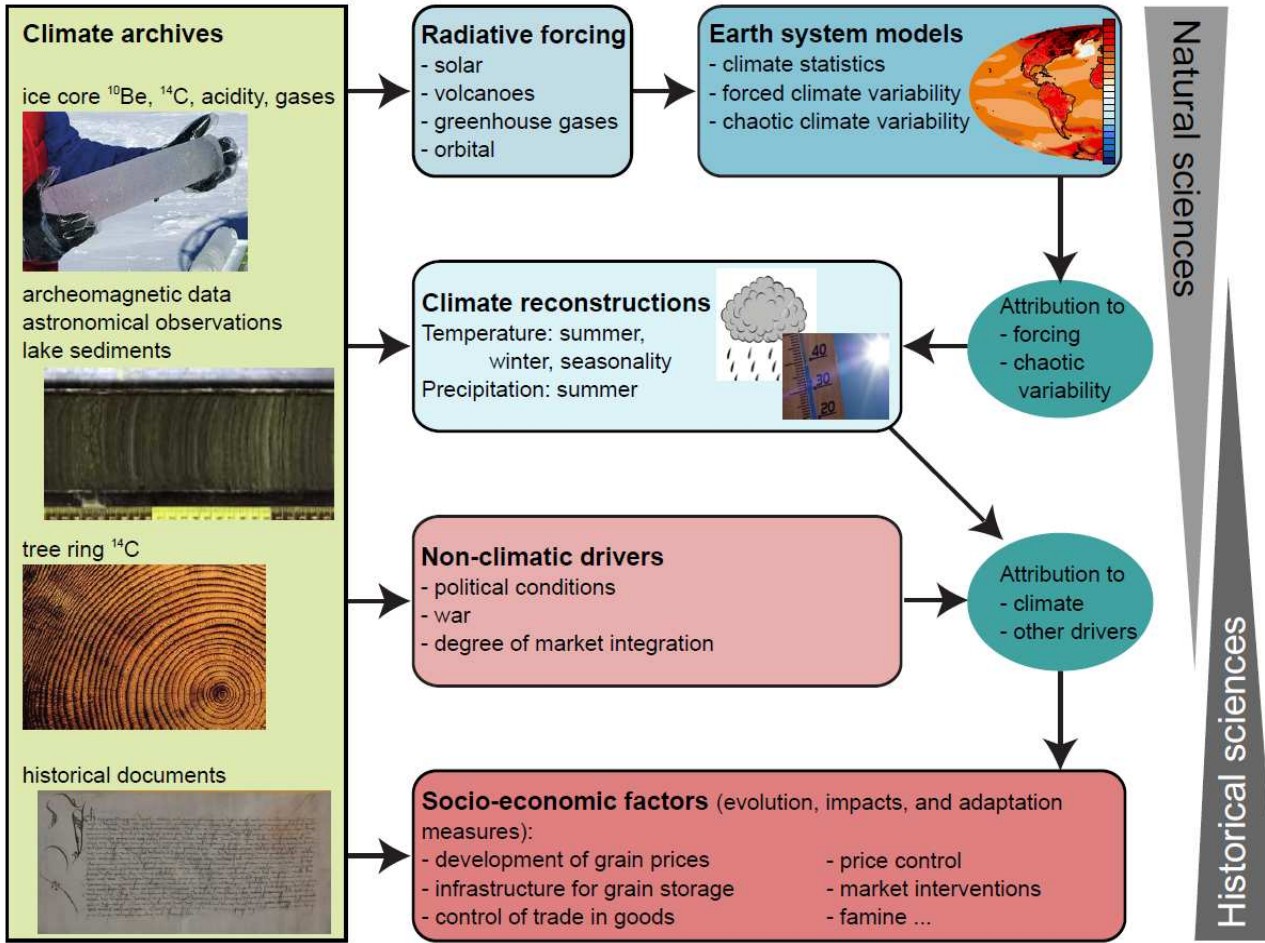

Fig. 1: Illustration of the research disciplines and methods brought together in this systematic assessment.

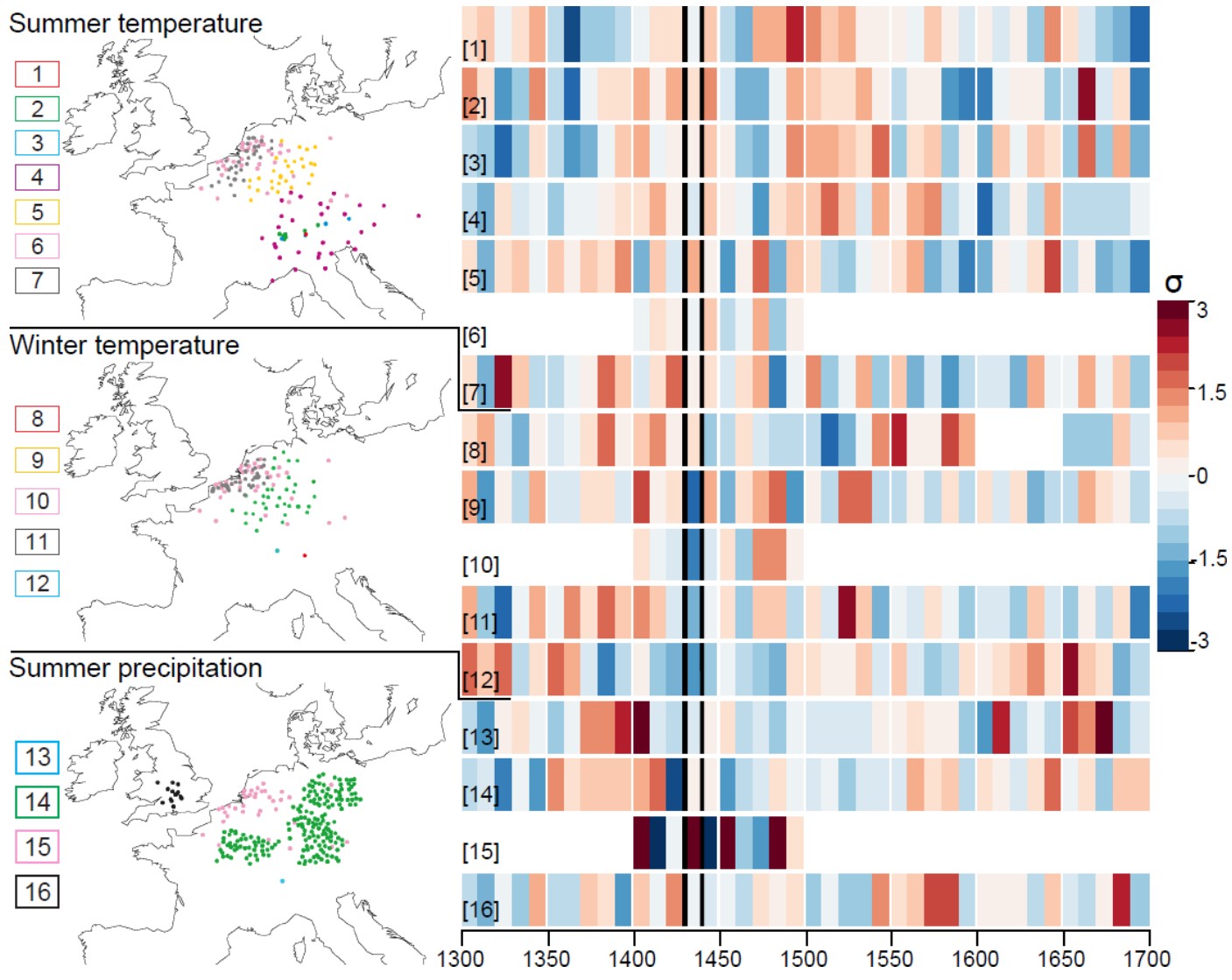

Fig. 2: Individual paleoclimate reconstructions for summer temperature, winter temperature and summer precipitation. [Left] Dots are specific sites considered by the different authors (listed from 1 to 16; Table 1). [Right] Decadal-scale (10-yr mean) summer temperature, winter temperature and summer precipitation for the 16 climate reconstructions, standardized with reference to the period 1300–1700 (data sets 6, 11 and 16: 1400–1500). The black lines enclose the decade 1430–1440. References: [1]Trachsel et al. (2010), [2]Büntgen et al. (2006), [3]Trachsel et al. (2012), [4,14]Büntgen et al. (2011), [5,9]Riemann et al. (2015), [6,10,15]Camenisch (2015b), [7,11]van Engelen et al. (2001), [8]de Jong et al. (2013), [12]Hasenfratz et al., in preparation, [13]Amann et al. (2015), [16]Wilson et al. (2013)

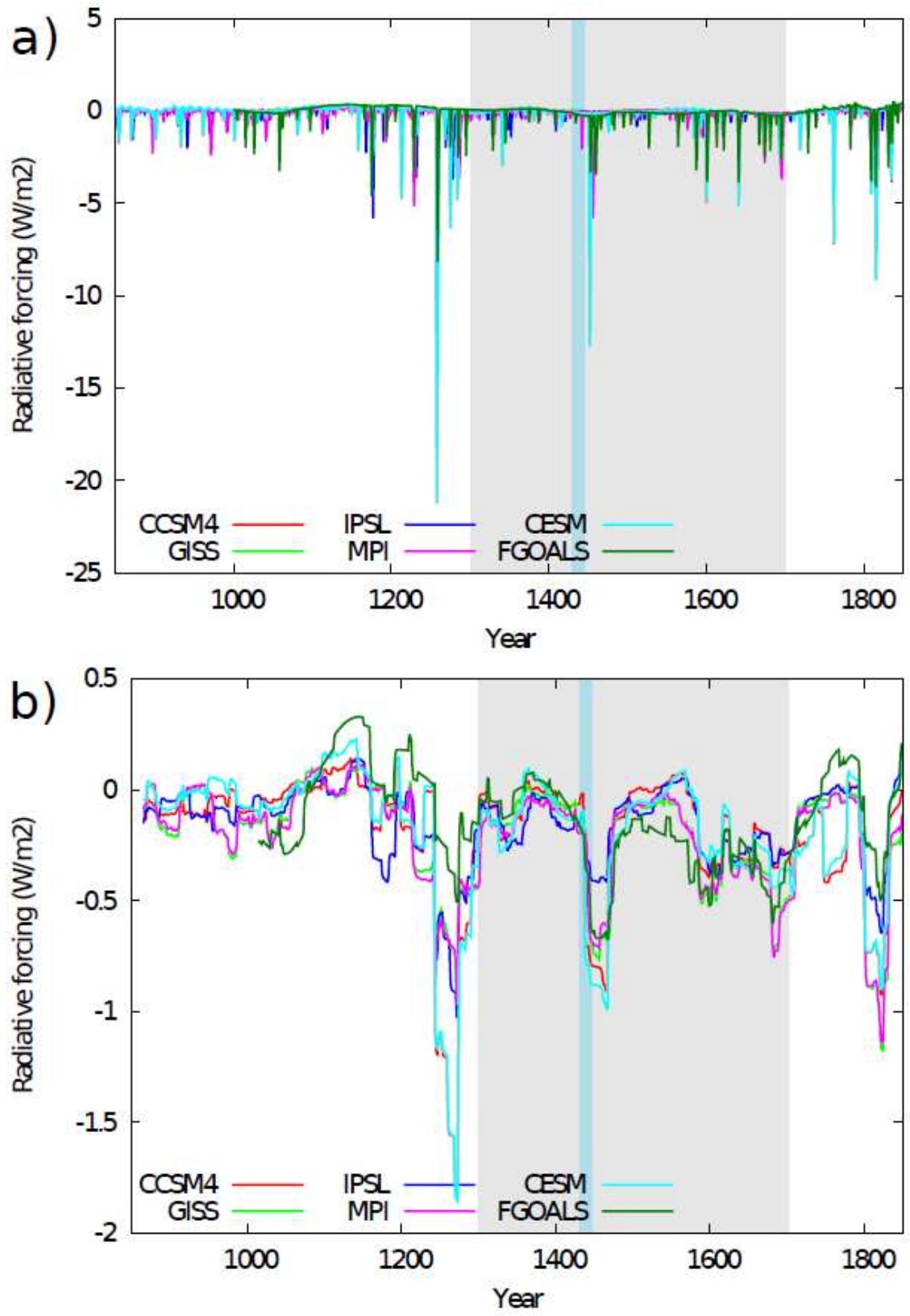

Fig. 3: (a) Estimations of total external forcings used by the models in Tab. 2 according to Fernández-Donado et al. (2013). The panel includes anomalies with respect to the period 1500–1850 including the contributions of anthropogenic (greenhouse gases and aerosols) and natural (solar variability and volcanic aerosols). (b) 31-year moving average filter outputs of (a).

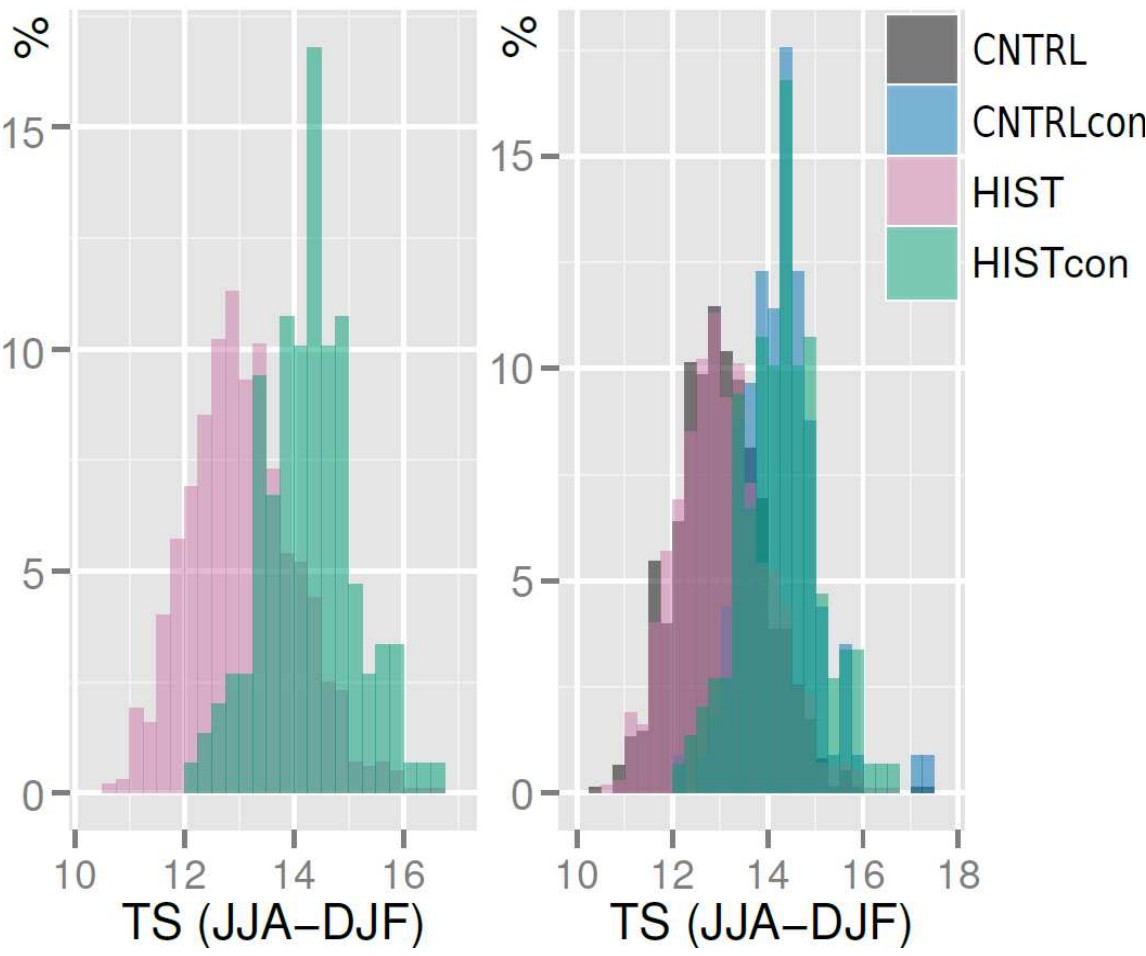

Fig. 4: Probability density functions of seasonality in surface temperature (TS; °C) averaged over Europe (8°W–22°E, 41–55°N) for all years and years with a very cold winter (conditional). The latter are defined by winter temperatures cooler than mean-1sigma. Left: for the years 850–1849 of the transient CESM simulation (HIST), right: also for the unforced control simulation (CNTRL; 600 yrs).

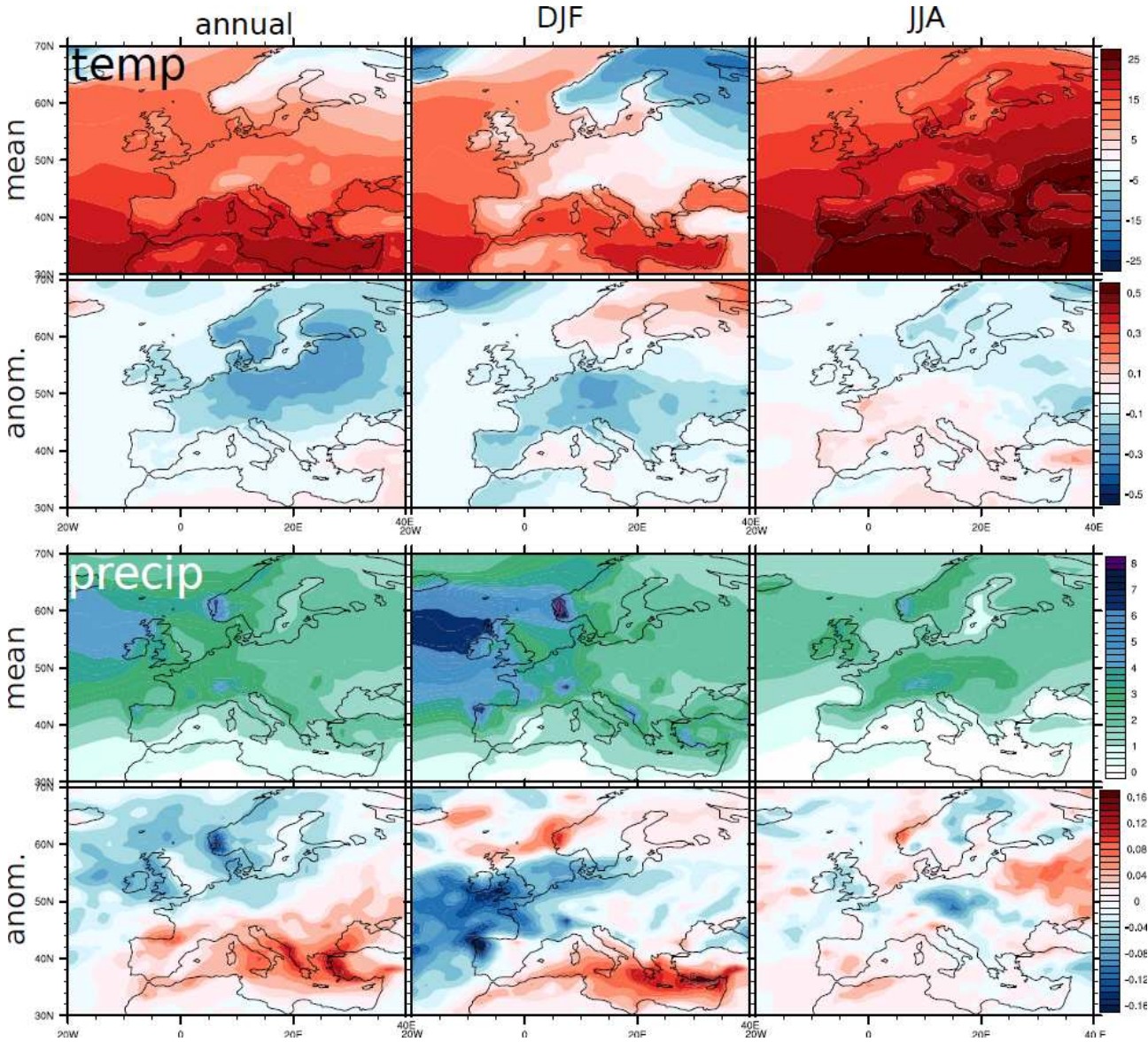

Fig. 5: Maps of surface temperature (top; °C) and precipitation (below; mm/day). Shown are annual (left), DJF (middle) and JJA (right) averages based on the transient simulation (years 850–1849) with CESM. First row: mean for all years. Second row: anomalies for years with strong seasonality in TS (> mean + 1sigma) compared to all years. Seasonality is defined as the difference between the means June–August and December–January of the respective year

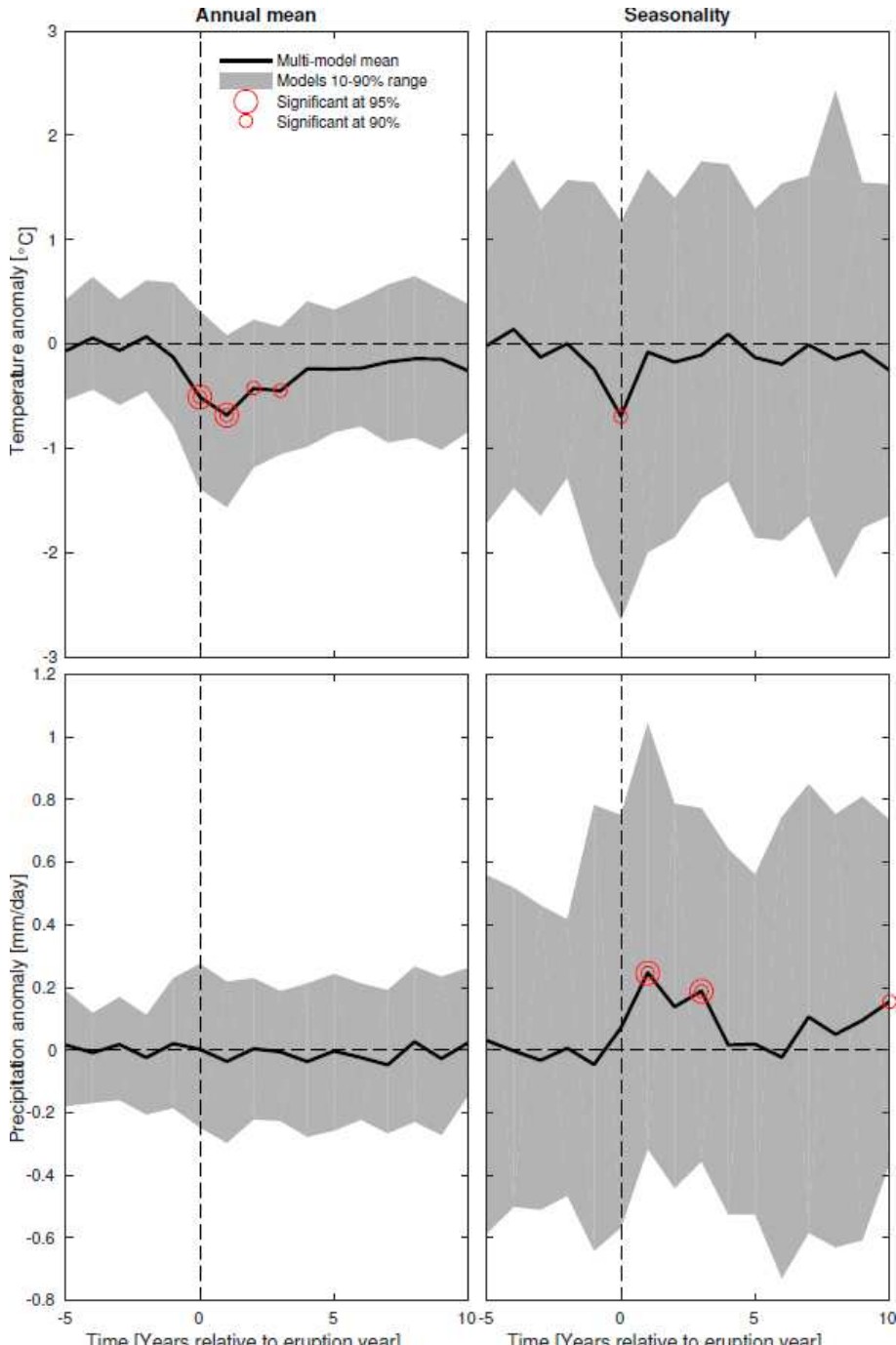

Fig. 6: Superposed Epoch Analysis on the ten strongest volcanic eruptions in six PMIP3 models, by measure of the respective forcing data set unit (aerosol optical depth or injection amount), over the period 850–1849. (Left) Annual mean temperature (top) and precipitation (bottom) and (right) seasonality defined as the difference between the means June–August and December–January of the respective year. Each of the 60 time series (6 models x 10 eruptions) is expressed as an anomaly to the mean of the five years preceding the eruption year (year 0). The shading indicates the 10–90% confidence interval, while the black solid line is the mean across all 60 time series. The red circles indicate significantly different means at 95% and 90% confidence according to a t-test comparing each year to the 5 year mean preceding the eruption year.

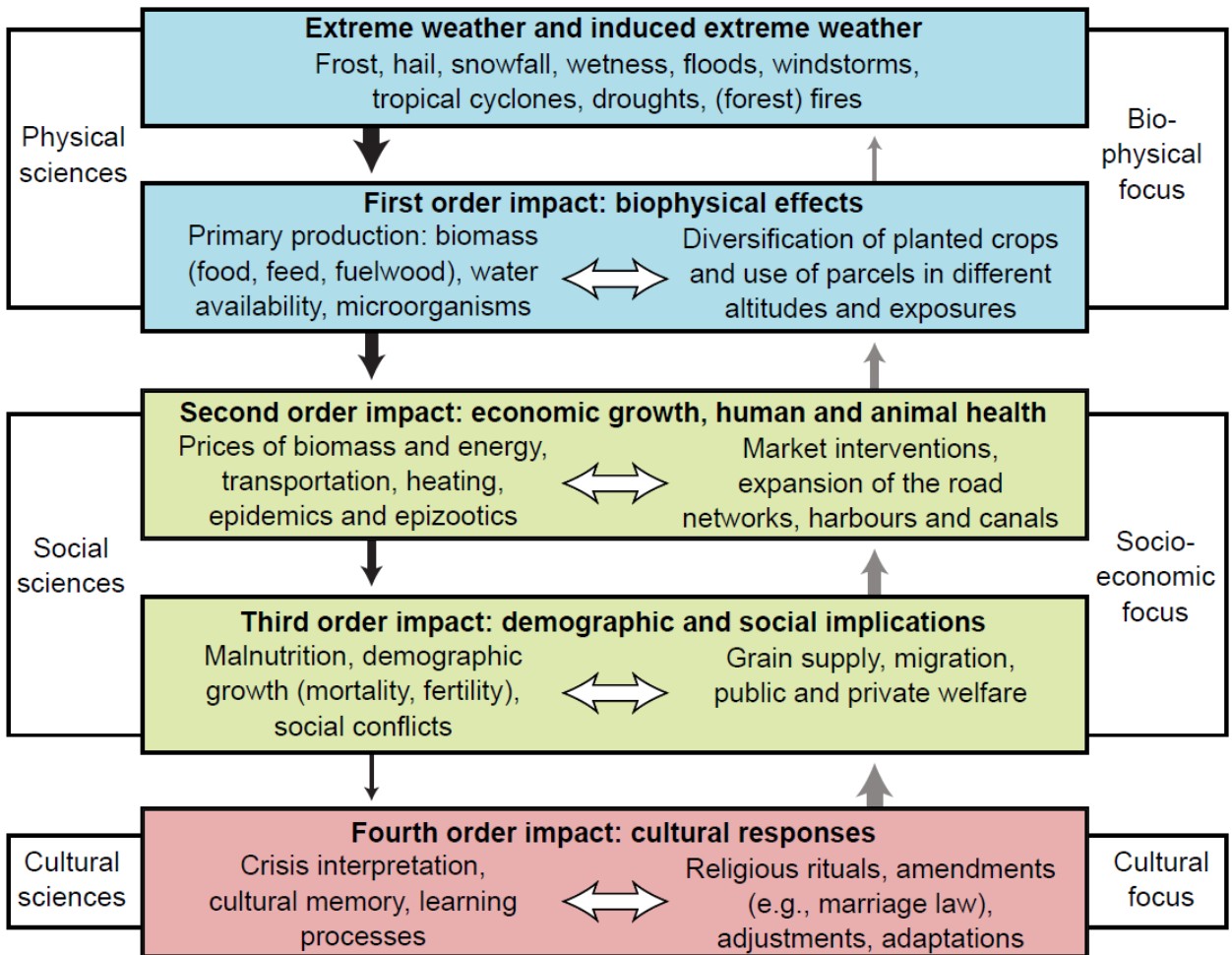

Fig. 7: This model developed by Daniel Krämer and Christian Pfister shows how climate interacts with society. Extreme weather causes biophysical effects on the first level, which can be followed by second order impacts that concern economic growth as well as human and animal health. On a third level are demographic and social implications situated whereas cultural responses act as fourth level impacts (Krämer, 2015; Luterbacher and Pfister, 2015).

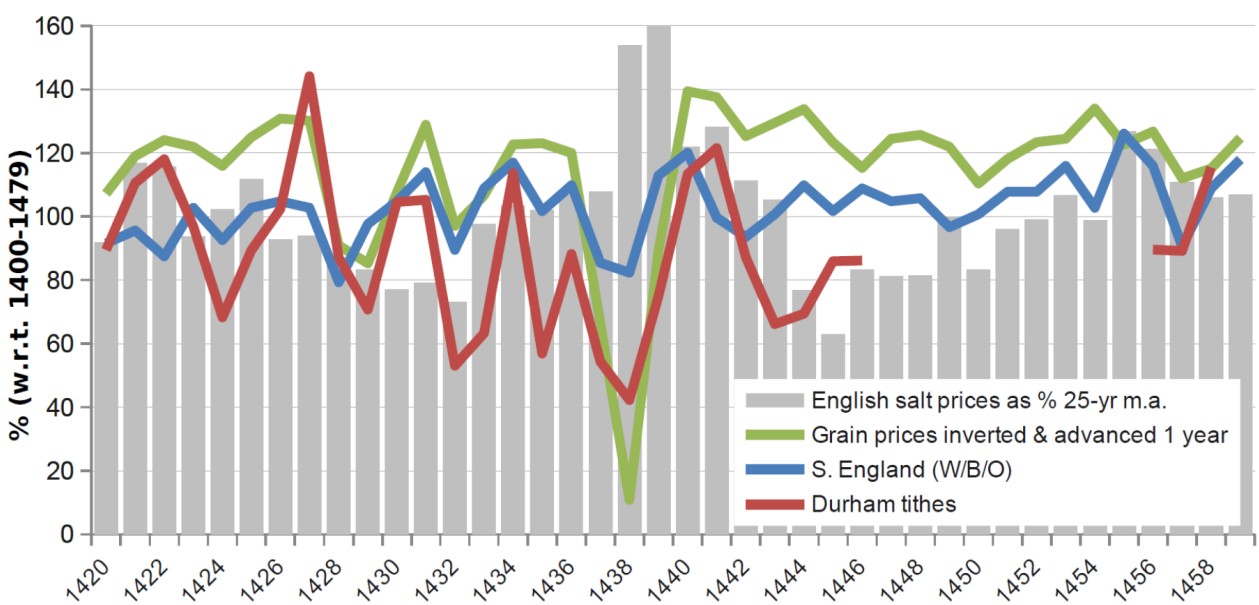

Fig. 8: Crop yields from Southern England (wheat, barley, oats), Durham tithes, English grain prices and English salt prices for the years 1420–1460 (values are given as anomalies with reference to (w.r.t.) the period 1400-1479). Shown are the years 1437 and 1438 in the back-to-back grain harvest failure in South England, the massive reduction in Durham tithe receipts and the marked inflation of grain prices for three consecutive years. In 1442, the harvests in South England are again poor. As far as the agricultural impacts of the Spörer Minimum are concerned in England, 1432–1442 stands out as the worst period, especially 1436–1438 (adapted from Campbell, 2012).