# Peer review of "The 1430s: A period of extraordinary internal climate variability during the early Spörer Minimum and its impacts in Northwestern and Central Europe"

_Climate of the Past, 2016_

## Short Comment (SC1) · 17 Feb 2016

I recommend that the authors reconsider their use of the term "early Spörer Minimum" as a name for a period of relatively cold climate.

The authors invent the term "early Spörer minimum" for the "cold period in Europe from around 1430 to 1445 CE". This clearly makes a link to the later period of low solar activity between 1460-1550, called the "Spörer Minimum" [1]. The "Spörer Minimum" is not a name for a period of cold (or warm/wet/dry/regional/global) climate change.

As the authors conclude that "...the increased occurrence of extremely cold winters during this decade can be attributed to unforced, internal variability and the resulting

atmospheric conditions.", the use of a name that is associated with a later period of low solar activity is inappropriate.

Even if a causal association was suggested, the use of a name from astronomy to describe a climate period is confusing. Different names should be used.

The authors also confuse the "Maunder Minimum", a name for another period of low solar activity [1], with a period of climate. e.g., On page 2 between lines 20-22, "While more recent cold events, such as ... the so-called Maunder Minimum in the 17th century..."

If the authors need to create a name for the climate period 1430 to 1445, they should consider a neutral name that does not imply a cause or exaggerate the climatic conditions at the time.

[1] Eddy, J.A., The Maunder Minimum, Science, 1976

———————————————

---

## Short Comment (SC2) · 24 Feb 2016

As mentioned in the document server of the university library, tambora.org is the successor of HisklidCORE (see https://www.freidok.uni-freiburg.de/proj/3 & https://www.freidok.uni-freiburg.de/data/10416)

So "HISKLID database" as mentioned in Tab.1 should be replaced by "tambora.org" as the data is available there. It has it's own DOI for citing: "10.6094/tambora.org".
* * *

---

## Referee Comment (RC1) · C. Lemmen (Referee) · 9 Mar 2016

**Summary**

The discussion paper titled "The early Spörer Minimum - a period of extraordinary climate and socio-economic changes in Western and Central Europe" submitted to Climate of the Past by Camenisch et al. aims to provide an end-to-end analysis of the interconnected histories of climate and society during a the winter-cold decade 1430-1440 in Northwestern and Central Europe. The authors provide evidence of climatic anomaly for this period based on multi-proxy and multi-site climate archives and from an ensemble of global climate simulations; they provide evidence on economic and social disturbances during and after this period based on documentary sources.

**Assessment**

This paper address relevant scientific questions within the scope of CP. It presents a novel combination of existing climate and historical data; bringing the historical socioeconomic data into a climate context and to the readership of CP is relevant. The conclusions drawn from the combination of the different data sources should be better substantiated, the manuscript could benefit from more careful statistical statements and rigorous application of significance level choice and evaluation. The literature cited is plentiful but appropriate, the abstract summarizes the paper well, but suffers from inconclusive statements; also the title should be improved. At several places, more information (that is in the SOM or "not shown") needs to be provided.

The presentation is well structured, the language is fluent, but it lacks precision. Mathematical symbols in the appendix are poorly typed and chosen. All figures are of insufficient quality for publication. I don't see much value in figs 1, 4b, 6, 7, and 9 in the main manuscript. The SOM is lacking a title and summary.

**Recommendation**

For scope, novelty and relevance I recommend to accept this paper. For accuracy, precision, language, figures, and conclusiveness the paper needs to be revised thoroughly.

**Detailed comments**

**Page 1 Title: as the short comment states, please reconsider your terminology with "Spörer Minimum" and early SPM; I would suggest also to find more distinct names for the period 1300 to 1700 versus the early part of this, esp 1430-1440. The geographical name is not correct; your region of investigation is rather Northwestern and Central than Western and Central Europe**

Authors: The list of authors is very long, although this study is not presenting original research but rather relies on data/simulations that have already been published. Why

is there such a long author list, and have all contributed to the manuscript? I suggest to move some to the acknowledgment section and clarify their contribution, if not directly related to research design, data production, analysis and writing of the paper.

l 38: not everyone might understand end-to-end, neither is your understanding necessarily the same as for people from other backgrounds. This should be defined.

**Page 2**

ll 4-6: consequence ... in order to be prepared. I did not see this chain of causation substantiated in the main text.

ll 7f: "Climate model simulations show .. internal variability" You demonstrate that it is *not* volcanic activity, and conclude then that this must be internal, but there could be other external forcings. (see below) Also, the climate model simulations look at cold winters, not that specific decade.

ll 8f: You do not test the hypothesis formally and don't conclude anything on this hypothesis later. I suggest to delete this sentence.

ll 12: "affected the socioeconomic system". Did all of them have effects, this is speculative

ll 12f: "cold resulted from forcing" imprecise; maybe the coldness can be attributed to forcing?

ll 14: "in the background" wrong "in"

ll 17: "include" and "inter alia" is redundant

ll 19: "negative" is a valuation, which should be avoided

ll 20: "all of which provoked", needs cautionary "may have", not all impacts provoke measures

ll 28f: "climatic and documentary" sources. These sources should not be connected by

"and", as one points to an analyis tool or discipline and the other to a material. Rather than "climatic archive" we have archives that record aspects of climate ...

ll 30: "devastating" avoid valuation

**Page 3**

ll 1: "more descriptive" is too general, be precise

ll 6: "punctuated" by

ll 8: "substantially" this is not substantiated or quantified

ll 9: "devastating" valuation

ll 13-15f: "low temperatures could destroy the winter seed" .. "combined with no or almost no snow cover". But your analysis does not include winter precipitation! This statement should have a reference.

ll 15 "usually had an effect". Citation needed.

ll 19 "*with* regard to"

ll 20 "and other" is too imprecise. The many societal changes that the paper later recounts should be mentioned. Otherwise, this sentence gives the impression that climate is the better understood, or more important driver of societal reactions.

ll 23 wording "replace crop failure", do you mean "recover from" or "mitigate"?

ll 29, 32f: "SPM was a period of rather cold" versus "only the Maunder Minimum was a coherent cold phase". This conflict should be resolved.

**Page 4**

ll 3: "major cooling happens after 1450" what is the relevance of this?

ll 3. "Taken at face value," colloquial

ll 4. "contradict" I don't see evidence for contradiction yet, since above, you argue that there are "other" impacts at work, too

ll 11: "dramatic change". Is not seen, it is one period of many with changes, has no long-lasting consequences

ll 28: "very moist early LIA". I don't see this in fig 2

ll 29: "context of entire millennium". Elaborate on Büntgen and Ammans findings and provide this context.

ll 30ff: "is rather diagnostic for" needs citation.

**Page 5**

ll 1: "Summers in the early SPM (1421–1450)". Previously, the early SPM was defined as 1431-1440 ???

ll 2: "see supporting". Where shown in supporting material?

ll 15-18: high seasonality could also indicate hot summer and normal winter, or medium cold winter and medium hot summer. Be clearer everywhere about your definition of high seasonality in combination with cold winter.

ll 19: "Figure 3 shows". I don't see this (see figure comments) well enough.

ll 20ff: Why do you show this in a figure if your data base is insufficient? A simple statement would be enough.

ll 31f: HIST and CNTRL are explained but not the "con" postfix used in the figure. Please elaborate on these four simulations

**Page 6 ll 2: "in land use" Which land use scenario do you use? In PMIP both the KK10 and HYDE are available and have very different land use forcings for your time period (see Fig 3a in Kaplan et al. 2011). A discussion on this should be included, also on the potential external forcing effect of land use (e.g. widespread deforestation by**

the 15th century in Central Europe). This may well be one of the "other" external (and anthropogenic) drivers of cold spells.

ll 6.: I see no relevance in the 31-year filtered data series. Remove.

ll 7: "There are large uncertainties". In what quantities? Are they related to natural variability, or to uncertainties in parameter estimation, or to subgrid or other not resolved processes?

ll 14: "models simulate an average decrease" provide reference

ll 30: reference fig S2; better: Show the ensemble average and range, and the CESM runs you analyse later for Tjja, Tdjf and seasonality in a new figure in the main text. leave S2 with the details of all models in the SOM

ll 32ff: Provide results from a statistical test on the significance of the observed differences.

ll 34: if 56% of years with large seasonality coincide with very cold winter, then 44 % should correspond to very warm summers. This seems not significant to me (but you could provide results from a test here.)

**Page 7**

ll 3: "external forcing does not affect modelled seasonality in Europe". Better: TSI, volcanic and GHG forcing do not affect ... (Land use is not considered, see discussion above). Can this statement be upheald without the statistical tests required earlier in this paragraph?

ll 6: "cold winter decades". It is not discussed how robust this analysis is, the figure S3 should be referenced, what is the motivation for the choice "5 out of 10" and "cooler than mean-1sigma" and the 600 year baseline? It is expected that in a normal distribution 2 out of 10 are cooler than 1 sigma, could you provide a more objective (reference!) analysis of the clustering of successively cold winters or many winters within a decade?

ll 8f: "years with strong seasonality show anomalously cold winters in Europe". By definition, as seasonality is not independent of cold winter temperature. Why show annual mean in Fig 6? Rather show seasonality. Why show at all, as the spatial finding (colder on land in winter) is easily report in text and the figure has not added value.

ll 17: "seasonality ... shows ... a reduction in seasonality after an eruption". No, in figure 7, the seasonality anomaly *after* the eruption is close to zero. Only *at* the eruption, seasonality is visibly (but is this significant) reduced. I don't see any significance in the seasonality trends shown in Fig 7, could you report this more objectively?

ll 21: "periods of frequent volcanic eruptions" You analysed not frequent eruptions, but the 10 strongest eruptions, without considering frequency ...

ll 24: "strongly" provide reference.

ll 25: "collapse" is a loaded word, considered biased by many historians, try to rephrase.

**page 8**

ll 1: "attributed" climate impacts cannot be attributed to a model!

ll 3: why "mainly"? What else?

ll 5: "Monocausal explanations are never sufficient" commonplace statement

ll 6: "reasons for these climate impacts" unclear. Reasons for the impact that climate had on society? or other reasons for impacts that are also interpreted as climatic?

ll 11: "could be drunk" citation needed.

ll 12: "winters during the 1430s" which ones exactly?

ll 14 "From 1424 to 1433 two flood and five drought years occurred," What is the long-term expected frequency of floods and droughts?

ll 20: "problems" too general. What kind?

ll 22f: "This temperature and precipitation pattern is also indicated by the models (see Fig. 6)". It is not valid to compare the coldest winters within the 600 year climate simulation with the historic winters in early SPM. Also, flood and precipitation is not comparable, neither are drought and temperature. Frequency and mean values are incomparable. This needs to be worked out in more detail, or Figure 6 can simply be removed.

ll 22 "the models". Only one model result is shown

ll 25 "several years" which ones precisely?

ll 34: "area of Berne" provide year.

**Page 9**

ll 8: the "North" might be unknown to non-English.

ll 9: what is "sweet wine"?

ll 16: cite Hungarian National Archives, not undefined acronym HNA

ll 19: "Europe's population sank to its lowest levels"

**Page 10**

ll 10: "organised" in what year?

ll 11: "Thus". Conjunction misused.

**Pag 11**

ll 6: "were known" are known?

ll 7: "as a reaction to the continuous rainfall" needs citation

ll 13: the uncapitalized word "gypsy" is wrong here, the capitalized version Gypsy associated with ethnic or racial slur and is to be avoided. Prefer "Romani", or at least use "then called" / "derogatorily termed" when using the word "Gypsy"

ll 14: "purely negative" is not precise and objective wording, rephrase.

ll 24f: "several wars aggravated the already tense situation". The authors make it sound, as if the social situation aggravated the root-causal climatic situation. I don't believe they want to convey this interpretation, but this needs to be sorted out. I believe that the authors want to say that climate stress aggravates the situation for already vulnerable societies. That way, the root cause is the social situation, and the trigger or aggravation comes from the climate impact (as clearly shown on page 12 ll 6).

**Page 12**

ll 17: "did not escalate into a demographic repeat" the demographic depression 1430 was the only one (as stated before by the authors). What is meant by this statement?

ll 24f: "Mutually reinforcing" sound like "circular arguments", maybe rephrase.

ll 25- next page: The relevance of herring for the period 1430-1440 is not demonstrated. The section is interesting but not relevant and could be removed.

**Page 13**

ll 15ff: "This strong increase in the seasonality of temperature suggests that, despite normal climatic conditions in the growing seasons, terrestrial ecosystem productivity was substantially decreased during this decade." This has not been conclusively shown.

ll 19: "caused by internal natural variability in the climate system rather than external forcing". This has not been shown, as land use was not considered and the evidence concentrated on ruling out volcanic forcing.

ll 27: "restrictions on the brewing of beer." This has not been shown

ll 32: " This period demonstrates .. a multitude of factors are needed". From the evidence presented, this cannot be concluded.

**Page 14**

ll 1: "threatening" is valuation, not objective.

ll 6: "Western", better Northwestern

ll 16: Style of equation poor. Choice of variable names, T1, T2 sxy nx, ny, syz non-standard. Typography not helpful for understanding. Use subscripts, proper roman/upright versus italic symbols, better weighted font sizes in equation. Avoid accent above x,y to denote mean (a bar would be more standard). A point is not a multiplication dot!

ll 17: Symbols don't match those in equation, see above.

ll 22: see above, poor equation style. zscores is not expleind

**Figure 1**

Not needed.

**Figure 2**

Graphical resolution of figure is poor. Please add (short) author information to legend, such that it is more easily seen when records from the same author appear in multiple panels. Adjust font to match the style of the journal

**Figure 3**

Most important figure, also requires major overhaul.

- adjust font to match the style of the journal - use consistent font sizes - add seasonality diagnostic to facilitate comparison - consider decadal moving average to provide smoother picture (and less visual clutter) - improve graphical representation of eSPM - annotate LIA and SPM periods - add markers for volcanic eruptions - add author and geographic information to legend - for 40 decades/observations, a p-value of 1-1/40 = .975 (i.e. approx 3 sigma) is appropriate to detect "anomalous" cold/hot periods (Thom-

son 1990), so extend color range to sigma = 3; the current choice highlights even small (1 sigma) excursions that are expected to be realized in 32% of the observations ....

Reconsider the entire figure with respect to the main conclusions the reader should draw from it. Give the important parts (1430-1440) more room and contrast, help by providing seasonality diagnostics.

**Figure 4**

- part a) could be moved to SOM, at better resolution and consistent font - part b) is irrelevant and should be removed

**Figure 5**

The rightmost panel is not legible. gray and blue (CTL) cannot be seen. There is a mismatch between the CTRL and CTL acronyms in figure and text. Postfix "con" is not explained (it should be conditional), PDF (probability density function) is not explained either.

- adjust font to match the style of the journal - use consistent font sizes and font colors - avoid duplication of data from HIST in rightmost panel

**Figure 6**

- not needed here, move to SOM - adjust font to match the style of the journal - use consistent font sizes and font colors - poor graphical resolution - abbreviations temp and precip are inconsistent with text - units are missing from color backgrounds - do not use divergent color map for temperature - consider relative anomaly instead of absolute anomaly - poor color choice in P anomaly: red (more) and blue (less) contrast with common perception of red (hot, dry) with blue (cold, wet); especially, since blue in precip mean plot indicates wetness.

**Figure 7**

- Move to SOM - adjust font to match the style of the journal - use consistent font

sizes and font colors - poor graphical resolution - motivate 10-90% confidence shading - highlight CESM results.

**Figure 9**

- adjust font to match the style of the journal - use consistent font sizes and font colors - poor graphical resolution

It is difficult to relate the dates to the values. The positioning of year labels along the x-axis is unclear; vertical grid lines would guide the eye, or choosing the bar plot for salt with small gaps between bars.

The y-scale is inappropriate. Percentage deviations should be presented on a logarithmic scale, such that doubling (200%) and halfing (50%) have equal distance to the reference. The reference should have a horizontal line.

The eSPM decade should be visually marked.

Extreme events could be annotated if historical accounts are provided by the authors.

**References**

Kaplan, J. O., Krumhardt, K. M., Ellis, E. C., Ruddiman, W. F., Lemmen, C., Goldewijk, K. K. (2011). Holocene carbon emissions as a result of anthropogenic land cover change. Holocene, 21(5).

Thomson, D. J. (1990). Time series analysis of Holocene climate data. Philosophical Transactions of the Royal Society of London A: Mathematical, Physical and Engineering Sciences, 330(1615), 601-616. ISO 690

---

## Referee Comment (RC2) · Anonymous Referee #2 · 10 Mar 2016

General Comments: Overall, I believe this article makes a valuable contribution to LIA European climate reconstruction and impacts, and I definitely recommend publication with revisions.

I am primarily a historian rather than climate scientist, and so I will focus mainly on the (climate) historical aspects of this paper.

1) Parts of the article, particularly the introduction, need very thorough editing, if not complete rewriting. There are simply too many phrases that could use correction and improvement to point them all out as a referee. Some ideas and conclusions are all but lost in poor prose. In places the language is too imprecise. For example, cooling "events" and "periods" seem to possibly cover everything from a year to centuries.

[Figure]

Since the author list includes many native speakers of English, I would expect more thorough editing for language.

2) The long-term demographic and economic context of Spörer Minimum societal impacts deserves even more emphasis. In the case of the 1310s and 1590s-1600s, issues of population pressure, diminishing land-holdings, and falling real wages have played a central role in explaining vulnerabilities and impacts. The case of the 1430s, when population was relatively low and real wages much higher, is very different. Indeed, I am surprised at the level of impacts described here, given that real wages in England for example were far higher in these years - despite the terrible weather - than during the early 14th and late 16th centuries.

Moreover, the bulk of Little Ice Age research has recently tended to emphasize runs of bad years related to volcanic eruptions and/or exceptional activity in the NAO in regard to climatic extremes and subsistence crises in Europe. In this case, we appear to have neither.

All of this leads to me wonder how Europeans of the 1430s still proved so vulnerable to bad weather and harvest failures. Without having a particular conclusion in mind, I would suggest that this puzzle might require further explanation. Perhaps the vulnerabilities created by conflicts (e.g., the Hundred Years War) deserve more emphasis. Or perhaps institutional factors, such as the lack of organized famine relief, exacerbated hunger and mortality. It may be that the demographic contraction of the period, while driving up wages, also created other vulnerabilities—a lower tax base, for instance, or higher transport costs, or fewer incentives to improve land and innovate in food production.

None of this is to say that I disagree with the authors' finding. Rather, I would suggest the authors have a chance here to make a larger contribution to our understanding of climatic vulnerabilities by emphasizing - and further explaining - the exceptional nature of this event. This discussion could go into the conclusion.

3) From the beginning of part 3 (p5, l25) onward, it is clear that one of the central features of this article is demonstrating how a period of apparently unforced internal variability could be quite extreme and bring significant impacts. This needs to be more clearly highlighted from the outset—rather than going unmentioned in the title and buried at the end of the abstract. I would even recommend retitling the article something like: "The 1430s: A Period of Extraordinary Internal Climate Variability and Its Impacts in Europe." Even starting with "The early Spörer Minimum" may prime the reader with thoughts of multi-decadal solar-driven cooling.

4) The descriptions of specific weather-related impacts in different parts of Europe might be easier to follow if the authors first offered a summary of relevant political, economic, and demographic factors (such as the long-term population decline since the Black Death, the Hundred Years War, etc.) rather than filling that information in later. It would help to set the stage, so to speak.

5) Are the authors suggesting that this event had any long-term consequences? Or was it just an unusually bad decade? In either case, how did the human perception and impact of the event compare to the more famous volcanic-induced cooling of the 1450s? In other words, did a climatic event arising from internal variability look and feel any different to Europeans than one driven by eruptions? The conclusion would be stronger if could address questions such as these.

6) In theory, the reader could look back and forth between figures 2 and 3 in order to figure out what climate anomalies occurred in each reconstructed region. In practice, however, that is difficult and time-consuming. I would prefer if the authors had a better way to visualize that data.

7) I do not mind the level of detail in section 4. However, parts of the text could be just as informative in half as many words. More importantly, it is not clear what the organizing principle of the section is. Sometimes it proceeds topically, sometimes geographically, sometimes chronologically (from one level of impact to the next). The text could use

some re-organization to make it more manageable.

8) I am disappointed that there is not a single historical quotation in section 4. If we want to know how people of the past perceived and experienced climate, it helps to hear their own words and narratives sometimes. Just a few well-chosen phrases would not only enliven the prose but also help establish what people of the time actually observed and regarded as noteworthy.

Specific Comments: >page 1, line 1: This statement is so vague that it could be misinterpreted in any number of ways.

>p2, l3: "normal but wet" appears contradictory; reword to indicate that summers had average temperatures but above-average precipitation. "strong seasonal cycle" is also vague and confusing (it's a phrase I would normally associate with something like the annual sales of winter coats)

>p2, l12: "affected the socio-economic systems" is so vague that it could be misinterpreted in any number of ways.

>p2, l14-15: This wording seems to conflate the gradual and increasing effects of orbital forcing with the short-term sporadic effects of eruptions.

>p.2, l25: "end-to-end assessment" appears to be a term of art in need of explanation

>p3, l3: does "Western Europe" here in include the Iberian Peninsula?

>p5, l6-7 and 13-14: I believe that Sigl et al. 2015 have updated this, and now assign the volcanic forcing of the 1590s-1600s to Huaynaputina and Nevado Ruiz (1595)—not Raung (1594). In any case, I believe the Nevado Ruiz eruption was the larger, and it is certainly well documented by contemporary witnesses.

>p5, l15-18: The article seems to have several poor explanations of winter temperature impacts on crops. It needs one good, well-placed explanation instead.

>p9, l13-14: The interactions between crop failures and plague deserve some brief

explanation.

>p11, l11-21: The authors may wish to compare the way Gypsies were blamed for bad weather in the Spörer Minimum to the way witches and Jews were blamed for weather and weather-related misfortunes in the late 16th-17th centuries (see Wolfgang Behringer, "Climatic Change and Witch Hunting: The Impact of the Little Ice Age on Mentalities," Climatic Change 43 (1999): 335–51; Dean Phillip Bell, "The Little Ice Age and the Jews: Environmental History and the Mercurial Nature of Jewish-Christian Relations in Early Modern Germany," AJS Review 32 (2008): 1–27.) It is interesting to see this transition, since it appears to confirm the thesis that Europeans transitioned from magical to demonic views of (weather) disasters—that is, from a belief that magical spells could cause bad weather to a belief that recourse to the devil or demons (as through witchcraft) could bring bad weather.

>Throughout the article, could the authors distinguish what measurement is used in "tree ring" data (i.e., ring width, density, or isotopes)?

>On page 33, figure 9, I don't see an explanation of the acronyms "w.r.t." or "W (B/O) YPS".

---

## Referee Comment (RC3) · Anonymous Referee #3 · 13 Mar 2016

General comments

The discussion paper titled "The early Spörer Minimum - a period of extraordinary climate and socio-economic changes in Western and Central Europe" is very stimulating and makes a valuable contribution to historical climatology. This article is undoubtedly within the scientific field of CP.

Based from climatic data and existing historical information, the major contribution and novelty of this paper is the systemic approach used to precisely measure the relationship between climate and societies for a short period of time (the 1430s) at a large spatial scale (mainly Central and Northwestern Europe) with a wide range of examples and comparisons. In itself, the constitution of an international team bringing together

most of the disciplines involved in climatic subjects is a recent and undeniably innovative approach.

The summary is concise and reflects the content of the article. The many references to previous researches are always relevant and systematically connected to the argumentation, so the contributions of the authors are evident. The overall presentation is clear and well structured, with a fluent and precise language.

I highly recommend publication of this article, after some revisions. As a historian, my comments are focused on this matter: I hope that it does not diminish the results of this inspiring paper.

1) The title reflects the content of the article but it should perhaps more clearly highlight its main contribution and its novelty, i.e. identify through a multifactorial approach the relationships between climate change and the social, economic, political, cultural and religious impacts in a short time scale. Somehow, this article aims to define or redefine the early Spörer Minimum from a Human point of view.

2) The summary speaks of "an end-to-end assessment": "a systemic (or systematic) survey" could perhaps be more comprehensive and better reflect the method used in the paper as illustrated in Fig. #1 and Fig. #8?

3) The relationship between famines and epidemics seem obvious and is mentioned several times in the paper (page 3, line 10; page 7, line 29; page 9, line 10-24; page 11, line 16). This matter is still debated, especially regarding the plague [Saluzzo, J.-F. : Des Hommes et des germes, Presses Universitaires de France, Paris, 2004 ; Audouin-Rouzeau, F. : Les Chemins de la peste. Le rat, la puce et l'homme, Presses Universitaires de Rennes, Rennes, 2003]. One or more localized examples emphasizing the chronological sequences famines-epidemics could be useful. Moreover, cases of ergotism (St Anthony's fire) are often perceived and described wrongly by the ancient as "epidemics", especially during subsistence crises.

4) The question of food trade requires some clarifications. Can we assume that international or interurban food trade (mentioned in page 10) is the standard in the fifteenth century? Or rather an exceptional measure during subsistence crisis? As shown in 1437 in Switzerland (page 10, lines 24-33), "municipal selfishness" may generally impeding the dilution of subsistence crisis by trade. For example, Flemish cities are opposed to the free movement of grains desired by the Duke of Burgundy in 1473 [Godard, J. : Dans les Pays-Bas bourguignons. Un conflit de politique commerciale, dans Annales d'histoire sociale, 1, 417-420, 1939].

5) Climate impacts on livestock are clearly exposed. A clarification may be useful: Did mass slaughter appear during the most intense crisis, contributing to aggravate the phenomenon? To what extent meat consumption is common in the first half of fifteenth century?

6) The construction of municipal grain storage capacities to avoid future subsistence crises is indicated in the summary (page 2, line 5) but deserves to be developed in the article. This is probably one of the most emblematic measures of adaptation of cities to climate change in the fifteenth century. These buildings often left their marks in the urban landscape till today. Several examples are directly related to the climatic context of 1430s: a "Kornhaus" was built in Cologne in 1439 and a second, much greater, was built from 1441, a "Kornhaus" was built in Strasbourg in 1441, etc.

7) Several (often extreme) cultural and religious responses are clearly discussed on page 11 but should perhaps be postponed towards the end of the article. Religious responses are a last resort after the failure of any other public policy (eg grain storage capacities, market interventions, etc.) to mitigate a subsistence crisis. The official violence (eg witch hunting, see Wolfgang Behringer) then becomes the last way to avoid the political crisis.

8) The fifteenth century context is complicated by the permanent state of war throughout much of Europe (pages 11-12). Any passing troops push regularly rural populations

to take refuge in cities with their food reserves, so that geopolitical stress may have (in an apparently paradoxical way) contributed to attenuate some subsistence crisis (at cities scale)?

9) If offshore fishing is suffering from climatic change due to displacements of migratory routes of fish stocks, does it helps to promote the development of inland freshwater fisheries? Or the development of fish farming (ponds, lakes)? When markets were unable to respond to the demand for fish during fasting (such as Lent), religious authorities may deliver exceptional authorizations to consume usually banned products (eggs, meat). Another form of (religious) adaptation to climate change?

As a conclusion, the many and very rich examples present in the paper show a massive implication of the civil or religious authorities at all spatial scales (states, cities) throughout Europe to mitigate or avoid subsistence crisis. Is it the same in earlier times? Otherwise, can we consider the 1430s as a matrix for subsequent crisis?

Specific comments and technical corrections

Page 2, line 3: "The particularly cold winters and normal but wet summers", maybe must it be specified that the summers are "normal" for temperatures?

Page 2, line 15: Go to the line before "The past climate is reconstructed from..." (change of subject)?

Page 3, lines 7-23: a paragraph to shift towards the end of the introduction (eg page 4, between lines 4 and 5) to better respect the plan of the article?

Section #4. Climate and weather impacts on the economy and society during the early Spörer Minimum: the structure of this section should be clarified by following more strictly the structure of Fig. # 8 or by displacing some paragraphs in a more linear order, eg 1) geopolitical context (the Hundred Years War, etc.) 2) extreme events and their impacts 3) grains 4) livestock 5) fishing 6) food trade 7) famines and plagues 8) public policies (grain storage, etc.) 9) religious and cultural responses.

[Figure]

**CPD**

Figures Fig. 1: "infrastructure for grain storage" rather to place in the "socio-economic factors" box? In the "socio-economic factors" box, add "control of trade in goods", "price control", "market interventions", etc.?

Fig. 2: For readability, do not separate the cards of their legends (table 1)?

Fig. 9 "W (B / A) YPS" –> clarify?

[Figure]

---

## Referee Comment (RC4) · Anonymous Referee #3 · 14 Mar 2016

- About the #8 point: I agree with you about the total absence of historical quotation.
- About specific comments: remarks very interesting and stimulating about the p.11, l.11-21, concerning the transition from magical to demonic views of weather disasters.

———————————————

---

## Author Comment (AC1) · 6 May 2016

Thank you for your comment. Please see the discussion at beginning of the reply to review 1.

---

## Author Comment (AC2) · 6 May 2016

Thank you for your comment.

This will be adjusted.
* * *

---

## Author Comment (AC3) · 6 May 2016

Thank you very much or your detailed review.

Main points:

Title: A point brought forward multiple times concerns the usage of the term "early Spörer Minimum", which also appears in the title. For clarification, we will improve the definition and usage of the different time periods in the introduction. Further, the title of the manuscript will be changed to "The 1430s: A period of extraordinary internal climate variability during the early Spörer Minimum and its impacts in Northwestern and Central Europe".

[Figure]

Geographical region: Following the reviewers, we will change this (also in the title) to "Northwestern and Central Europe".

- Below we answer the comments point by point.

Review 1

The discussion paper titled "The early Spörer Minimum - a period of extraordinary climate and socio-economic changes in Western and Central Europe" submitted to Climate of the Past by Camenisch et al. aims to provide an end-to-end analysis of the interconnected histories of climate and society during the winter-cold decade 1430-1440 in Northwestern and Central Europe. The authors provide evidence of climatic anomaly for this period based on multi-proxy and multi-site climate archives and from an ensemble of global climate simulations; they provide evidence on economic and social disturbances during and after this period based on documentary sources. # Assessment This paper address relevant scientific questions within the scope of CP. It presents a novel combination of existing climate and historical data; bringing the historical socioeconomic data into a climate context and to the readership of CP is relevant. The conclusions drawn from the combination of the different data sources should be better substantiated, the manuscript could benefit from more careful statistical statements and rigorous application of significance level choice and evaluation. The literature cited is plentiful but appropriate, the abstract summarizes the paper well, but suffers from inconclusive statements; also the title should be improved. At several places, more information (that is in the SOM or "not shown") needs to be provided. The presentation is well structured, the language is fluent, but it lacks precision. Mathematical symbols in the appendix are poorly typed and chosen. All figures are of insufficient quality for publication. I don't see much value in figs 1, 4b, 6, 7, and 9 in the main manuscript. The SOM is lacking a title and summary.

- The technical shortcomings listed will be clarified.

**Recommendation For scope, novelty and relevance I recommend to accept this paper. For accuracy, precision, language, figures, and conclusiveness the paper needs to be revised thoroughly.**

- The quality of the figures suffered during the copying into the word file which was then uploaded to Copernicus. The quality will be better in the final version.

**Detailed comments ## Page 1 Title: as the short comment states, please reconsider your terminology with "Spörer Minimum" and early SPM; I would suggest also to find more distinct names for the period 1300 to 1700 versus the early part of this, esp1430-1440. The geographical name is not correct; your region of investigation is rather Northwestern and Central than Western and Central Europe Authors: The list of authors is very long, although this study is not presenting original research but rather relies on data/simulations that have already been published. Why is there such a long author list, and have all contributed to the manuscript? I suggest to move some to the acknowledgment section and clarify their contribution, if not directly related to research design, data production, analysis and writing of the paper.**

- Title: See above. We agree that the list of authors is very long but all of them contributed substantially in the one or the other way to the paper. This is why we will not change it.

l 38: not everyone might understand end-to-end, neither is your understanding necessarily the same as for people from other backgrounds. This should be defined.

- End-to-end will be replaced with systematic.

**Page 2 ll 4-6: consequence ... in order to be prepared. I did not see this chain of causation substantiated in the main text.**

- We will improve this part regarding the granaries in the main text.

ll 7f: "Climate model simulations show .. internal variability" You demonstrate that it is *not* volcanic activity, and conclude then that this must be internal, but there could be other external forcings. (see below) Also, the climate model simulations look at cold

winters, not that specific decade.

- We assume that the reviewer hints at land use changes. See below (p.6/l.2). Still note that the model simulations are fully coupled, consequently each simulation develops its own internal variability which is not necessarily in lockstep with reality. Consequently, it is not meaningful to focus on that specific decade.

ll 8f: You do not test the hypothesis formally and don't conclude anything on this hypothesis later. I suggest to delete this sentence.

- Hypothesised will be replaced by suggested.

ll 12: "affected the socioeconomic system". Did all of them have effects, this is speculative

- we will add "might have".

ll 12f: "cold resulted from forcing" imprecise; maybe the coldness can be attributed to forcing?

- "Resulted from" will be changed to "can be attributed to".

ll 14: "in the background" wrong "in"

- Will be corrected to "against the background of".

ll 17: "include" and "inter alia" is redundant

- we will delete "inter alia"

ll 19: "negative" is a valuation, which should be avoided

- ok

ll 20: "all of which provoked", needs cautionary "may have", not all impacts provoke measures

- Will be replaced by "many of which provoked".

ll 28f: "climatic and documentary" sources. These sources should not be connected by "and", as one points to an analysis tool or discipline and the other to a material. Rather than "climatic archive" we have archives that record aspects of climate ...

- we will change that

ll 30: "devastating" avoid valuation

- Will be deleted.

**Page 3**

ll 1: "more descriptive" is too general, be precise

- we will change that into "descriptions"

ll 6: "punctuated" by

- Will be changed.

ll 8: "substantially" this is not substantiated or quantified

- Will be deleted.

ll 9: "devastating" valuation

- Will be deleted.

ll 13-15f: "low temperatures could destroy the winter seed" .. "combined with no or almost no snow cover". But your analysis does not include winter precipitation! This statement should have a reference.

- We will add the reference (Pfister 1999) and emphasise that only temperature was analysed.

ll 15 "usually had an effect". Citation needed.

- It is the same citation as in the next sentences (Camenisch 2015b). This is why we

did not repeat it. If necessary, we will add it.

ll 19 "*with* regard to"

- Done.

ll 20 "and other" is too imprecise. The many societal changes that the paper later recounts should be mentioned. Otherwise, this sentence gives the impression that climate is the better understood, or more important driver of societal reactions.

- Examples will be added.

ll 23 wording "replace crop failure", do you mean "recover from" or "mitigate"?

- mitigate

ll 29, 32f: "SPM was a period of rather cold" versus "only the Maunder Minimum was a coherent cold phase". This conflict should be resolved.

- Clarified to "only the Maunder Minimum was a globally coherent cold phase".

**Page 4**

ll 3: "major cooling happens after 1450" what is the relevance of this?

- It illustrates that the models are able to reproduce the climatic conditions following strong volcanic eruptions.

ll 3. "Taken at face value," colloquial.

- Will be deleted.

ll 4. "contradict" I don't see evidence for contradiction yet, since above, you argue that there are "other" impacts at work, too

- Will be rewritten

ll 11: "dramatic change". Is not seen, it is one period of many with changes, has no

long-lasting consequences

- Will be rephrased.

ll 28: "very moist early LIA". I don't see this in fig 2 ll 29: "context of entire millennium". Elaborate on Büntgen and Ammans findings and provide this context.

- Sentence will be deleted

ll 30ff: "is rather diagnostic for" needs citation.

- changed to 'variability illustrates (cumulative) volcanic forcing or internal (unforced) variability'

**Page 5**

ll 1: "Summers in the early SPM (1421–1450)". Previously, the early SPM was defined as 1431-1440 ???

- Will be clarified in the introduction

ll 2: "see supporting". Where shown in supporting material?

- Shown in: Fig. S1 (Reconstructed European summer temperature anomalies (w.r.t. 1961-1990) for the years 1430-1439. For further information, see Euro-Med2k Consortium, in press)

ll 15-18: high seasonality could also indicate hot summer and normal winter, or medium cold winter and medium hot summer. Be clearer everywhere about your definition of high seasonality in combination with cold winter.

- Agreed. The respective definition is given in the figure captions; we will try to make this clearer.

ll 19: "Figure 3 shows". I don't see this (see figure comments) well enough.

- Since we have both summer- and winter time series for only very few locations, we

refrained from showing the seasonality.

ll 20ff: Why do you show this in a figure if your data base is insufficient? A simple statement would be enough.

- The data base is insufficient to draw strong conclusions. But it is still valuable information, offering an idea of the hydroclimatic conditions during the investigated period.

ll 31f: HIST and CNTRL are explained but not the "con" postfix used in the figure. Please elaborate on these four simulations

- The "con" in "conditional" is now underlined. Details about the simulations are outside the scope of the paper and can be found in the stated publications (Keller et al., 2015; Lehner et al., 2015)

**Page 6**

ll 2: "in land use" Which land use scenario do you use? In PMIP both the KK10 and HYDE are available and have very different land use forcings for your time period (see Fig 3a in Kaplan et al. 2011). A discussion on this should be included, also on the potential external forcing effect of land use (e.g. widespread deforestation by the 15th century in Central Europe). This may well be one of the "other" external (and anthropogenic) drivers of cold spells.

- In the simulation we use the reconstuction by Pongratz et al. (2008) and Hurtt et al. (2011) which are merged in the year 1500 AD. During the 15th century, land use change in Northwestern and Central Europe is a very gradual and steady process. There is no indication that these changes could have implications on the climatic conditions on such large spatial scales as investigated here.

ll 6.: I see no relevance in the 31-year filtered data series. Remove.

- Due to the strong volcanic forcing the minima in solar forcing are hardly visible in the regular time series. Therefore, we prefer to leave it.

ll 7: "There are large uncertainties". In what quantities? Are they related to natural variability, or to uncertainties in parameter estimation, or to subgrid or other not resolved processes? - Will be rewritten.

ll 14: "models simulate an average decrease" provide reference

- This refers to the analysis conducted here, consequently there is no reference. Will be clarified.

ll 30: reference fig S2; better: Show the ensemble average and range, and the CESM runs you analyse later for Tjja, Tdjf and seasonality in a new figure in the main text. leave S2 with the details of all models in the SOM.

- Maybe there was a misunderstanding. Sentence will be clarified.

ll 32ff: Provide results from a statistical test on the significance of the observed differences. ll 34: if 56% of years with large seasonality coincide with very cold winter, then 44 % should correspond to very warm summers. This seems not significant to me (but you could provide results from a test here.) # Page 7 ll 3: "external forcing does not affect modelled seasonality in Europe". Better: TSI, volcanic and GHG forcing do not affect ... (Land use is not considered, see discussion above). Can this statement be upheald without the statistical tests required earlier in this paragraph?

- Our argumentation builds on the fact that unforced control simulations (without external forcing) and transient simulations (with external forcing) show the same/comparable signals. These signals are not necessarily statistically significant which, however, does not affect the main message. Land use: see above.

ll 6: "cold winter decades". It is not discussed how robust this analysis is, the figure S3 should be referenced, what is the motivation for the choice "5 out of 10" and "cooler than mean-1sigma" and the 600 year baseline? It is expected that in a normal distribution 2 out of 10 are cooler than 1 sigma, could you provide a more objective (reference!) analysis of the clustering of successively cold winters or many winters within a decade?

- Will be clarified.

ll 8f: "years with strong seasonality show anomalously cold winters in Europe". By definition, as seasonality is not independent of cold winter temperature. Why show annual mean in Fig 6? Rather show seasonality. Why show at all, as the spatial finding (colder on land in winter) is easily report in text and the figure has not added value.

- Years with strong seasonality do not necessarily result from cold winters, they could also originate from extremely warm summers in combination with average winters. The figure illustrates that – at least in this specific model – this is not the case. Further, annual values are shown to give the reader an impression of the model performance.

ll 17: "seasonality ... shows ... a reduction in seasonality after an eruption". No, in figure 7, the seasonality anomaly *after* the eruption is close to zero. Only *at* the eruption, seasonality is visibly (but is this significant) reduced. I don't see any significance in the seasonality trends shown in Fig 7, could you report this more objectively?

- Correct the response is in the year of eruption – will be changed accordingly. We will check the statistical significance.

ll 21: "periods of frequent volcanic eruptions" You analysed not frequent eruptions, but the 10 strongest eruptions, without considering frequency . . .

- As the response to the strong volcanic eruption is only found in the year of the eruption a first order estimation is that in periods of frequent strong volcanic eruptions just multiple repeats this response and this leads to a period of low seasonality. Will be clarified.

ll 24: "strongly" provide reference.

- Will be added.

ll 25: "collapse" is a loaded word, considered biased by many historians, try to rephrase.

- Ok.

**page 8**

ll 1: "attributed" climate impacts cannot be attributed to a model!

- Will be rephrased.

ll 3: why "mainly"? What else?

- There are also younger sources which tell the events from a later epoch (e. g. second half of 15th or 16th century). Possibly, those sources are not as reliable as the contemporary sources.

ll 5: "Monocausal explanations are never sufficient" commonplace statement.

- Will be deleted.

ll 6: "reasons for these climate impacts" unclear. Reasons for the impact that climate had on society? or other reasons for impacts that are also interpreted as climatic?

- Will be rephrased.

ll 11: "could be drunk" citation needed.

- Dawson 2009, as in the subsequent sentence.

ll 12: "winters during the 1430s" which ones exactly?

- We well give more precise information.

ll 14 "From 1424 to 1433 two flood and five drought years occurred," What is the long-term expected frequency of floods and droughts?

- We will try to add more information but maybe this would go too much into details.

ll 20: "problems" too general. What kind?

- Will be rephrased.

ll 22f: "This temperature and precipitation pattern is also indicated by the models (see Fig. 6)". It is not valid to compare the coldest winters within the 600 year climate simulation with the historic winters in early SPM. Also, flood and precipitation is not comparable, neither are drought and temperature. Frequency and mean values are incomparable. This needs to be worked out in more detail, or Figure 6 can simply be removed.

- Will be rephrased.

ll 22 "the models". Only one model result is shown

- We delete the plural.

ll 25 "several years" which ones precisely?

- Will be added where possible and reasonable.

ll 34: "area of Berne" provide year.

- Ok.

**Page 9**

ll 8: the "North" might be unknown to non-English.

- Of England. Should be clear from the context.

ll 9: what is "sweet wine"?

- A certain type of wine. Should be clear from the context.

ll 16: cite Hungarian National Archives, not undefined acronym HNA

- Ok.

ll 19: "Europe's population sank to its lowest levels"

- Will be rephrased.

**Page 10**

ll 10: "organised" in what year?

- This is rather too detailed and not the scope of the paper.

ll 11: "Thus". Conjunction misused.

- Will be rephrased.

**Pag 11**

ll 6: "were known" are known?

- Will be rephrased.

ll 7: "as a reaction to the continuous rainfall" needs citation

- The entire paragraph is Bauch 2015.

ll 13: the uncapitalized word "gypsy" is wrong here, the capitalized version Gypsy associated with ethnic or racial slur and is to be avoided. Prefer "Romani", or at least use "then called" / "derogatorily termed" when using the word "Gypsy" ll 14: "purely negative" is not precise and objective wording, rephrase.

- Yes of course. We apologize for this mistake. It happened while we were shortening the text. The author of this paragraph provided a correct version.

ll 24f: "several wars aggravated the already tense situation". The authors make it sound, as if the social situation aggravated the root-causal climatic situation. I don't believe they want to convey this interpretation, but this needs to be sorted out. I believe that the authors want to say that climate stress aggravates the situation for already vulnerable societies. That way, the root cause is the social situation, and the trigger or aggravation comes from the climate impact (as clearly shown on page 12 ll 6).

- Will be rephrased.

**Page 12**

ll 17: "did not escalate into a demographic repeat" the demographic depression 1430 was the only one (as stated before by the authors). What is meant by this statement?

- The famine of 1316-1322 was an extremely severe demographic catastrophe and historians, especially medievalist would compare any famine to that one of 1316-1322. Will be clarified.

ll 24f: "Mutually reinforcing" sound like "circular arguments", maybe rephrase.

- Will be rephrased.

ll 25- next page: The relevance of herring for the period 1430-1440 is not demonstrated. The section is interesting but not relevant and could be removed.

- The paragraph provides a different view on the investigated time period. We prefer to leave it, but will be rephrased.

**Page 13**

ll 15ff: "This strong increase in the seasonality of temperature suggests that, despite normal climatic conditions in the growing seasons, terrestrial ecosystem productivity was substantially decreased during this decade." This has not been conclusively shown.

- Will be rephrased.

ll 19: "caused by internal natural variability in the climate system rather than external forcing". This has not been shown, as land use was not considered and the evidence concentrated on ruling out volcanic forcing.

- Will be rephrased.

ll 27: "restrictions on the brewing of beer." This has not been shown

- Will be added to the main text.

ll 32: " This period demonstrates .. a multitude of factors are needed". From the evidence presented, this cannot be concluded.

- Will be rephrased.

**Page 14**

ll 1: "threatening" is valuation, not objective.

- Will be deleted.

ll 6: "Western", better Northwestern

- Ok.

ll 16: Style of equation poor. Choice of variable names, T1, T2 sxy nx, ny, syz non-standard. Typography not helpful for understanding. Use subscripts, proper roman/upright versus italic symbols, better weighted font sizes in equation. Avoid accent above x,y to denote mean (a bar would be more standard). A point is not a multiplication dot! ll 17: Symbols don't match those in equation, see above. ll 22: see above, poor equation style. zscores is not expleind

- Will be adjusted.

**Figure 1 Not needed.**

- This figure keeps together the different parts of the paper. It shows how the results of different disciplines are linked together in the question of the examined period. We prefer to leave the figure in the paper.

**Figure 2 Graphical resolution of figure is poor. Please add (short) author information to legend, such that it is more easily seen when records from the same author appear in multiple panels. Adjust font to match the style of the journal**

- Figures 2 and 3 will be combined, the author information added to the caption.

**Figure 3 Most important figure, also requires major overhaul. -adjust font to match**

the style of the journal -use consistent font sizes -add seasonality diagnostic to facilitate comparison -consider decadal moving average to provide smoother picture (and less visual clutter) -improve graphical representation of eSPM -annotate LIA and SPM periods -add markers for volcanic eruptions -add author and geographic information to legend -for 40 decades/observations, a p-value of 1-1/40 = .975 (i.e. approx 3 sigma) is appropriate to detect "anomalous" cold/hot periods (Thomson 1990), so extend color range to sigma = 3; the current choice highlights even small (1 sigma) excursions that are expected to be realized in 32% of the observations .... Reconsider the entire figure with respect to the main conclusions the reader should draw from it. Give the important parts (1430-1440) more room and contrast, help by providing seasonality diagnostics.

- Figures 2 and 3 will be combined, the author information added to the caption. The color range will be changed to 3 sigma. Since we have both summer- and winter time series for only very few locations, we refrained from showing the seasonality diagnostic. To keep the focus on the 1430s, and to preserve the overall clarity and readability of the figure, we do not annotate LIA, SPM and specific volcanic eruptions (the latter are difficult since we show decadal means).

**Figure 4 -part a) could be moved to SOM, at better resolution and consistent font -part b) is irrelevant and should be removed**

- We prefer to keep the figure as it is; see above.

**Figure 5 The rightmost panel is not legible. gray and blue (CTL) cannot be seen. There is a mismatch between the CTRL and CTL acronyms in figure and text. Postfix "con" is not explained (it should be conditional), PDF (probability density function) is not explained either. -adjust font to match the style of the journal -use consistent font sizes and font colors -avoid duplication of data from HIST in rightmost panel**

- Acronyms/prefixes will be adjusted; PDF will be explained. Our argumentation builds on the fact that unforced control simulations (without external forcing) and transient simulations (with external forcing) show the same/comparable signals. This is illustrated

by the left panel; we prefer to leave it like it is.

**Figure 6 -not needed here, move to SOM -adjust font to match the style of the journal -use consistent font sizes and font colors -poor graphical resolution -abbreviations temp and precip are inconsistent with text -units are missing from color backgrounds -do not use divergent color map for temperature -consider relative anomaly instead of absolute anomaly -poor color choice in P anomaly: red (more) and blue (less) contrast with common perception of red (hot, dry) with blue (cold, wet); especially, since blue in precip mean plot indicates wetness.**

- Will be adjusted. The figure is an illustration of the potential climatic conditions during the investigated time period. We prefer to not move it to the SOM.

**Figure 7 -Move to SOM -adjust font to match the style of the journal -use consistent font sizes and font colors -poor graphical resolution -motivate 10-90% confidence shading -highlight CESM results.**

- Will be adjusted. We prefer to not move it to the SOM

**Figure 9 -adjust font to match the style of the journal -use consistent font sizes and font colors -poor graphical resolution It is difficult to relate the dates to the values. The positioning of year labels along the x-axis is unclear; vertical grid lines would guide the eye, or choosing the bar plot for salt with small gaps between bars. The y-scale is inappropriate. Percentage deviations should be presented on a logarithmic scale, such that doubling (200%) and halfing (50%) have equal distance to the reference. The reference should have a horizontal line. The eSPM decade should be visually marked. Extreme events could be annotated if historical accounts are provided by the authors.**

- Will be adjusted.

---

## Author Comment (AC4) · 6 May 2016

Review 2

- Thank you very much or your detailed review.

General Comments: Overall, I believe this article makes a valuable contribution to LIA European climate reconstruction and impacts, and I definitely recommend publication with revisions. I am primarily a historian rather than climate scientist, and so I will focus mainly on the (climate) historical aspects of this paper.

1) Parts of the article, particularly the introduction, need very thorough editing, if not complete rewriting. There are simply too many phrases that could use correction and

improvement to point them all out as a referee. Some ideas and conclusions are all but lost in poor prose. In places the language is too imprecise. For example, cooling "events" and "periods" seem to possibly cover everything from a year to centuries. Since the author list includes many native speakers of English, I would expect more thorough editing for language.

- Okay, we will work on that and use more precise expressions.

2) The long-term demographic and economic context of Spörer Minimum societal impacts deserves even more emphasis. In the case of the 1310s and 1590s-1600s, issues of population pressure, diminishing land-holdings, and falling real wages have played a central role in explaining vulnerabilities and impacts. The case of the 1430s, when population was relatively low and real wages much higher, is very different. Indeed, I am surprised at the level of impacts described here, given that real wages in England for example were far higher in these years - despite the terrible weather – than during the early 14th and late 16th centuries. Moreover, the bulk of Little Ice Age research has recently tended to emphasize runs of bad years related to volcanic eruptions and/or exceptional activity in the NAO in regard to climatic extremes and subsistence crises in Europe. In this case, we appear to have neither. All of this leads to me wonder how Europeans of the 1430s still proved so vulnerable to bad weather and harvest failures. Without having a particular conclusion in mind, I would suggest that this puzzle might require further explanation. Perhaps the vulnerabilities created by conflicts (e.g., the Hundred Years War) deserve more emphasis. Or perhaps institutional factors, such as the lack of organized famine relief, exacerbated hunger and mortality. It may be that the demographic contraction of the period, while driving up wages, also created other vulnerabilities⅘AĚĞTa lower tax base, for instance, or higher transport costs, or fewer incentives to improve land and innovate in food production. None of this is to say that I disagree with the authors' finding. Rather, I would suggest the authors have a chance here to make a larger contribution to our understanding of climatic vulnerabilities by emphasizing - and further explaining - the exceptional nature of

this event. This discussion could go into the conclusion.

- Thank you for this remark. We will try to improve that part of the paper by adding further information and by rephrasing a number of paragraphs.

3) From the beginning of part 3 (p5, l25) onward, it is clear that one of the central features of this article is demonstrating how a period of apparently unforced internal variability could be quite extreme and bring significant impacts. This needs to be more clearly highlighted from the outset⅘AĚĞTrather than going unmentioned in the title and buried at the end of the abstract. I would even recommend retitling the article something like: "The 1430s: A Period of Extraordinary Internal Climate Variability and Its Impacts in Europe." Even starting with "The early Spörer Minimum" may prime the reader with thoughts of multi-decadal solar-driven cooling.

- See discussion in the beginning of the reply to review 1.

4) The descriptions of specific weather-related impacts in different parts of Europe might be easier to follow if the authors first offered a summary of relevant political, economic, and demographic factors (such as the long-term population decline since the Black Death, the Hundred Years War, etc.) rather than filling that information in later. It would help to set the stage, so to speak.

- A short paragraph will be added.

5) Are the authors suggesting that this event had any long-term consequences? Or was it just an unusually bad decade? In either case, how did the human perception and impact of the event compare to the more famous volcanic-induced cooling of the 1450s? In other words, did a climatic event arising from internal variability look and feel any different to Europeans than one driven by eruptions? The conclusion would be stronger if could address questions such as these.

- A short paragraph will be added.

6) In theory, the reader could look back and forth between figures 2 and 3 in order to

figure out what climate anomalies occurred in each reconstructed region. In practice, however, that is difficult and time-consuming. I would prefer if the authors had a better way to visualize that data.

- Figures 2 and 3 will be combined, the author information added to the caption.

7) I do not mind the level of detail in section 4. However, parts of the text could be just as informative in half as many words. More importantly, it is not clear what the organizing principle of the section is. Sometimes it proceeds topically, sometimes geographically, sometimes chronologically (from one level of impact to the next). The text could use some re-organization to make it more manageable.

- Section 4 is structured following Krämer 2015 / Pfister&Luterbacher 2015 (see Fig. 8).

8) I am disappointed that there is not a single historical quotation in section 4. If we want to know how people of the past perceived and experienced climate, it helps to hear their own words and narratives sometimes. Just a few well-chosen phrases would not only enliven the prose but also help establish what people of the time actually observed and regarded as noteworthy.

- Yes of course we will add some quotation.

Specific Comments: >page 1, line 1: This statement is so vague that it could be misinterpreted in any number of ways.

- Will be rephrased.

>p2, l3: "normal but wet" appears contradictory; reword to indicate that summers had average temperatures but above-average precipitation. "strong seasonal cycle" is also vague and confusing (it's a phrase I would normally associate with something like the annual sales of winter coats)

- Will be rephrased.

>p2, l12: "affected the socio-economic systems" is so vague that it could be misinterpreted in any number of ways.

- Will be rephrased.

>p2, l14-15: This wording seems to conflate the gradual and increasing effects of orbital forcing with the short-term sporadic effects of eruptions.

- Will be rephrased.

>p.2, l25: "end-to-end assessment" appears to be a term of art in need of explanation

- End-to-end will be replaced with systematic.

>p3, l3: does "Western Europe" here in include the Iberian Peninsula? No.

- We will replace western by north-western

>p5, l6-7 and 13-14: I believe that Sigl et al. 2015 have updated this, and now assign the volcanic forcing of the 1590s-1600s to Huaynaputina and Nevado Ruiz (1595)⅘AĚĞ Tnot Raung (1594). In any case, I believe the Nevado Ruiz eruption was the larger, and it is certainly well documented by contemporary witnesses.

- Will be updated.

>p5, l15-18: The article seems to have several poor explanations of winter temperature impacts on crops. It needs one good, well-placed explanation instead.

- Will be improved.

>p9, l13-14: The interactions between crop failures and plague deserve some brief explanation.

- Will be added.

>p11, l11-21: The authors may wish to compare the way Gypsies were blamed for bad weather in the Spörer Minimum to the way witches and Jews were blamed for

weather and weather-related misfortunes in the late 16th-17th centuries (see Wolfgang Behringer, "Climatic Change and Witch Hunting: The Impact of the Little Ice Age on Mentalities," Climatic Change 43 (1999): 335–51; Dean Phillip Bell, "The Little Ice Age and the Jews: Environmental History and the Mercurial Nature of Jewish-Christian Relations in Early Modern Germany," AJS Review 32 (2008): 1–27.) It is interesting to see this transition, since it appears to confirm the thesis that Europeans transitioned from magical to demonic views of (weather) disastersâĔŸAĔĞ Tthat is, from a belief that magical spells could cause bad weather to a belief that recourse to the devil or demons (as through witchcraft) could bring bad weather.

- Thank you, will be added.

>Throughout the article, could the authors distinguish what measurement is used in "tree ring" data (i.e., ring width, density, or isotopes)?

- Not the scope of the paper, the information is available in the provided references.

>On page 33, figure 9, I don't see an explanation of the acronyms "w.r.t." or "W (B/O) YPS".

- Will be adjusted.

―――――――――――――――――――

---

## Author Comment (AC5) · 6 May 2016

Review 3

- Thank you very much or your detailed review.

1) The title reflects the content of the article but it should perhaps more clearly highlight its main contribution and its novelty, i.e. identify through a multifactorial approach the relationships between climate change and the social, economic, political, cultural and religious impacts in a short time scale. Somehow, this article aims to define or redefine the early Spörer Minimum from a Human point of view.

- See discussion in the beginning of the reply to review 1.

[Figure]

2) The summary speaks of "an end-to-end assessment": "a systemic (or systematic) survey" could perhaps be more comprehensive and better reflect the method used in the paper as illustrated in Fig. #1 and Fig. #8?

- Will be rephrased.

3) The relationship between famines and epidemics seem obvious and is mentioned several times in the paper (page 3, line 10; page 7, line 29; page 9, line 10-24; page 11, line 16). This matter is still debated, especially regarding the plague [Saluzzo, J.-F. : Des Hommes et des germes, Presses Universitaires de France, Paris, 2004; Audouin-Rouzeau, F. : Les Chemins de la peste. Le rat, la puce et l'homme, Presses Universitaires de Rennes, Rennes, 2003]. One or more localized examples emphasizing the chronological sequences famines-epidemics could be useful. Moreover, cases of ergotism (St Anthony's fire) are often perceived and described wrongly by the ancient as "epidemics", especially during subsistence crises.

- Thank you, we will add a paragraph on this and also show the debate on that topic including ergotism.

4) The question of food trade requires some clarifications. Can we assume that international or interurban food trade (mentioned in page 10) is the standard in the fifteenth century? Or rather an exceptional measure during subsistence crisis? As shown in 1437 in Switzerland (page 10, lines 24-33), "municipal selfishness" may generally impeding the dilution of subsistence crisis by trade. For example, Flemish cities are opposed to the free movement of grains desired by the Duke of Burgundy in 1473 [Godard, J. : Dans les Pays-Bas bourguignons. Un conflit de politique commerciale, dans Annales d'histoire sociale, 1, 417-420, 1939].

- A short paragraph on grain trade will be added. This topic is well examined for the 15th century.

5) Climate impacts on livestock are clearly exposed. A clarification may be useful: Did

mass slaughter appear during the most intense crisis, contributing to aggravate the phenomenon? To what extent meat consumption is common in the first half of fifteenth century?

- We will check the question of mass slaughter and add a few sentences on the meat consumption (which is also still debated).

6) The construction of municipal grain storage capacities to avoid future subsistence crises is indicated in the summary (page 2, line 5) but deserves to be developed in the article. This is probably one of the most emblematic measures of adaptation of cities to climate change in the fifteenth century. These buildings often left their marks in the urban landscape till today. Several examples are directly related to the climatic context of 1430s: a "Kornhaus" was built in Cologne in 1439 and a second, much greater, was built from 1441, a "Kornhaus" was built in Strasbourg in 1441, etc.

- Thank you, we will improve that part.

7) Several (often extreme) cultural and religious responses are clearly discussed on page 11 but should perhaps be postponed towards the end of the article. Religious responses are a last resort after the failure of any other public policy (eg grain storage capacities, market interventions, etc.) to mitigate a subsistence crisis. The official violence (eg witch hunting, see Wolfgang Behringer) then becomes the last way to avoid the political crisis.

- Thank you, we will improve this.

8) The fifteenth century context is complicated by the permanent state of war throughout much of Europe (pages 11-12). Any passing troops push regularly rural populations to take refuge in cities with their food reserves, so that geopolitical stress may have (in an apparently paradoxical way) contributed to attenuate some subsistence crisis (at cities scale)?

- Thank you, we will improve this.

9) If offshore fishing is suffering from climatic change due to displacements of migratory routes of fish stocks, does it helps to promote the development of inland freshwater fisheries? Or the development of fish farming (ponds, lakes)? When markets were unable to respond to the demand for fish during fasting (such as Lent), religious authorities may deliver exceptional authorizations to consume usually banned products (eggs, meat). Another form of (religious) adaptation to climate change? As a conclusion, the many and very rich examples present in the paper show a massive implication of the civil or religious authorities at all spatial scales (states, cities) throughout Europe to mitigate or avoid subsistence crisis. Is it the same in earlier times? Otherwise, can we consider the 1430s as a matrix for subsequent crisis?

- We will check this.

Specific comments and technical corrections

Page 2, line 3: "The particularly cold winters and normal but wet summers", maybe must it be specified that the summers are "normal" for temperatures?

- Will be rephrased.

Page 2, line 15: Go to the line before "The past climate is reconstructed from..." (change of subject)?

- We will consider this.

Page 3, lines 7-23: a paragraph to shift towards the end of the introduction (eg page 4, between lines 4 and 5) to better respect the plan of the article?

- We will consider this.

Section #4. Climate and weather impacts on the economy and society during the early Spörer Minimum: the structure of this section should be clarified by following more strictly the structure of Fig. # 8 or by displacing some paragraphs in a more linear order, eg 1) geopolitical context (the Hundred Years War, etc.) 2) extreme events and

their impacts 3) grains 4) livestock 5) fishing 6) food trade 7) famines and plagues 8) public policies (grain storage, etc.) 9) religious and cultural responses.

- We will consider this. The structure of this part will be improved anyway (see reply to review 2).

Figures Fig. 1: "infrastructure for grain storage" rather to place in the "socio-economic factors" box? In the "socio-economic factors" box, add "control of trade in goods", "price control", "market interventions", etc.?

- We will check this.

Fig. 2: For readability, do not separate the cards of their legends (table 1)?

- Figures 2 and 3 will be combined, the author information added to the caption.

Fig. 9 "W (B / A) YPS" –> clarify?

- Will be added.

---

## Author Response (AR2)

Thank you very much for your detailed review and the very useful suggestions.

TITLE
The new title "The 1430s: A period of extraordinary internal climate variability during the early Spörer Minimum and its impacts in Northwestern and Central Europe" does not work gramatically. It would work, for example, if "and its" was replaced with "with". Also, the "impacts" should be qualified in oder to reflect the special aspect of this paper. "Societal impacts" or "socioeconomic impacts" or "impacts on societies" or something like that. Could otherwise be anything, such as impacts on ecosystems, erosion, or endless other things readers of Climate of the Past might think of.

We changed the title into: "The 1430s: A cold period of extraordinary internal climate variability during the early Spörer Minimum with social and economic impacts in Northwestern and Central Europe"

ABSTRACT
The abstract is not well organised but jumping back an forth between different aspects of the paper, i.e. the climatic and the impacts analysis. Also, the Spörer minimum is first referred to, then later explained. And it falls short of making the main points of the paper. Please improve. This is particularly important if the article is proposed for highlighting, which might be an option, given the interesting climate-society connection.
After revision of the manuscript the abstract content will probably need to be adapted accordingly

We restructured the abstract and distinguished between aspects of climatic analysis and impact analysis. The term Spörer is deleted.

LANGUAGE
As mentioned by the reviewers, the language is often imprecise and a little awkward. A revised version that is acceptable needs to be improved in that aspect. I put some edits into the manuscript, partly to demonstrate the kind of issues meant. The entire manuscript should be combed through, please. I am sure there is enough capacity for that in the author team.

We thoroughly went through the whole manuscript and clarified/homogenized the language.

TEXT ORGANISATION
The text organisation should be improved, as also pointed out by a reviewer, but then mostly disregarded in the revision. The text deficiencies are also surprising given the prominent list of co-authors and cumulative publishing experience.
The text still reads as three separate parts of different character. I appreciate that it might be challenging to homogenise texts of researchers with different cultures, but am asking nevertheless for some further, moderate improvements:
- The introduction chapter lacks clarity what the purpose of the paper is. Moreover, from about page 3 line 29 a lot reads like preempting the results/discussion already. Please check to make a clear distinction between introduction/framing etc, and presentation of the results.
- Chapters 2 and 3 provide information that goes significantly beyond the key aspect of the paper (according to the new title), i.e. the 1430's. We can accept that as background information, but I am asking for a summary paragraph at the end of the data and modelling chapters that highlight the most relevant conclusions for the key storyline.

We changed the text organisation of the abstract, the introduction and the conclusion. In particular the parts of the introduction which contained results were integrated into the respective chapters. Chapter two and three end now with a brief summary of the major results.

- Chapter 4 (as mentioned by a reviewer) is simply disproportionately long - too long to be read by CP readers in its current form. I suggest to cut where possible (some suggestion in my annotated text, but please check for further potential for condensing) AND to move climate information to chapter 2 even if from historical sources AND add sub-chapter structure to chapter 4. Sub-chapters could logically follow the scheme in figure 7.

All climate information from chapter 4 has been removed to chapter 2. Chapter 4 has been shortened considerably. Subtitles have been added to the chapter.

One referee also suggested to highlight adaptation measures and to separate out the more societal/political from the religious ones. I think this would also be a great way of presenting the adaptation topic better, given that we have major adaptation challenges in front of us today also.

The paper falls short of translating the results to what we can conclude for our modern society and current situation about things like vulnerability and adaptation responses. It would be great of the authors could expand a bit on that, if they dare. Could be in the conclusions or as a short (sub-)chapter before.

We added a paragraph to the conclusion in regard to vulnerability and adaptation.

Please check the use of the term weather. In several places, it seems to be used inappropriately when climate is actually meant.

Done

A major finding highlighted from the paleoclimate analysis is the attribution of the high-seasonality period to internal variability, not external forcing. It would be good to explain (probably mostly in chapter 3) why this is relevant for this paper, as opposed to just detecting the seasonality and then looking at the impacts on societies.

This finding might not be of particular importance for the impact part, but we think that it is also important to address the cause behind these particular climatic conditions (especially in the framework of such an interdisciplinary effort).

CONCLUSIONS
As asked by referees, the conclusions should be better substantiated (see also some annotations I made into the text).
We changed the structure and added information.

FIGURES
Fig. 2 has turned out very nicely. Other figures were criticised by referees to be less important, and I agree with them. On the other hand, as you seem to feel strongly to keep them, I am fine to keep them in the paper, even if they might illustrate non-central points.
Please note a few remarks in the annotated manuscript.

Done

[revised manuscript text omitted]